# "It's because I think too much": Perspectives and experiences of adults with hypertension engaged in HIV care in northern Tanzania

Preeti Manavalan[1,2,3]*, Linda Minja[4], Lisa Wanda[4], Julian T. Hertz[3,5], Nathan M. Thielman[2,3], Nwora Lance Okeke[2], Blandina T. Mmbaga[3,4,6], Melissa H. Watt[3,7]

**1** Division of Infectious Diseases and Global Medicine, Department of Medicine, University of Florida, Gainesville, FL, United States of America, **2** Division of Infectious Diseases, Department of Medicine, Duke University Medical Center, Durham, NC, United States of America, **3** Duke Global Health Institute, Durham, NC, United States of America, **4** Kilimanjaro Clinical Research Institute, Moshi, Tanzania, **5** Division of Emergency Medicine, Department of Surgery, Duke University Medical Center, Durham, NC, United States of America, **6** Kilimanjaro Christian Medical University College, Moshi, Tanzania, **7** Department of Population Health Sciences, University of Utah, Salt Lake City, UT, United States of America

* preeti.manavalan@gmail.com

## Abstract

### Background

Hypertension, a leading risk for cardiovascular mortality, is an important co-morbidity among people living with HIV (PLHIV). In Tanzania, hypertension prevalence among PLHIV approaches 20 to 30%. However, most patients are unaware of their diagnosis and are not receiving treatment. Understanding the barriers to hypertension care is a critical first step in developing interventions to improve cardiovascular outcomes among PLHIV in Tanzania and similar settings.

### Methods

Between September 1st and November 26th, 2018 thirteen semi structured in-depth interviews were conducted with hypertensive patients engaged in HIV care in two HIV clinics located in government health facilities in northern Tanzania. Interviews were audio-recorded, translated into English, transcribed and thematically coded using NVivo. Data analysis was conducted using applied thematic analysis.

### Results

Participants had a median age of 54 (IQR 41–65) years. Of the 13 participants, eight stated they had used antihypertensive medication previously, but only one participant described current use of antihypertensive therapy. All participants were currently using antiretroviral therapy. The data revealed a range of themes including limited hypertension knowledge. Universally, all participants believed that "thinking too much", i.e. stress, was the major contributor to hypertension and that by "reducing thoughts", one may control hypertension. Additional emerging themes included a perceived overlap between hypertension and HIV, delays in hypertension diagnosis and linkage to care, challenges with provider communication and counseling, reluctance towards antihypertensive medication, lack of

**Data Availability Statement:** Due to the small sample size and the detailed content of the interview transcripts, including sensitive and potentially identifying information, it is not possible

to de-identify the qualitative data. The ethics committee has not approved public release of this type of data, and the data supporting the findings of this study will be available upon reasonable request. Data requests can be sent to the corresponding author or to the Duke University Health System IRB with protocol number Pro00091126, Mailing Address: 2424 Erwin Road, Suite 405, Durham, NC, Tel: (919) 668-5111, Email: eIRB@mc.duke.edu.

**Funding:** PM received support for this study from the US National Institutes of Health Fogarty International Center, grant number D43TW009337, [https://www.fic.nih.gov/Grants/Search/Pages/fellows-scholars-D43TW009337.aspx] and from the National Institutes of Health Ruth L. Kirschstein National Research Service Award, grant number 5T32AI007392, [https://www.niaid.nih.gov/grants-contracts/training-grants]. MHW received support from the Duke Center for AIDS Research, grant number P30AI064518, [https://cfar.duke.edu/].The funders had no role in study design, data collection and analysis, decision to publish, or preparation of the manuscript.

**Competing interests:** The authors have declared that no competing interests exist.

integration of hypertension and HIV care, and additional structural barriers to hypertension care.

## Conclusions

Participants described multiple, intersecting challenges related to hypertension management. Barriers specific to PLHIV included siloed care, HIV-related stigma, and burden from multiple medical conditions. Multifaceted strategies that seek to address structural barriers, hypertension education, psychosocial stressors and stigma, and that are integrated within HIV care are urgently needed to improve cardiovascular outcomes among PLHIV in sub-Saharan Africa.

## Introduction

With scale-up of antiretroviral therapy (ART), HIV-related mortality has declined rapidly [1]. People living with HIV (PLHIV) are now living longer and are faced with a double burden of disease: HIV infection and chronic non-communicable diseases (NCDs), including cardiovascular disease (CVD) [2, 3]. Compared to the general population, PLHIV have more than a 50% increased risk of CVD [4, 5]. Hypertension is a major risk factor for CVD and is the leading risk for death world-wide [6], and it is estimated that up to a third of adults with treated HIV meet criteria for hypertension [7].

Sub-Saharan Africa (SSA) has the highest prevalence of hypertension in the world [8], and in Tanzania approximately 1 in 3 people may be hypertensive [9]. Despite this increased burden of disease, most patients with hypertension in SSA are not on treatment [10]. These concerning trends are also present among PLHIV [11]: a recent study in northern Tanzania found that among a cohort of hypertensive patients enrolled in HIV care, 65% were unaware of their diagnosis, 90% were not on antihypertensive treatment and all had uncontrolled blood pressure [12].

Several studies have identified barriers to hypertension care among the general patient population in SSA. Impediments to care include lack of access to blood pressure equipment and drugs, absence of guidelines for hypertension management, shortages in clinical staff, gaps in provider and patient knowledge, and financial constraints [13–15]. Whether PLHIV face unique barriers to hypertension care remains unclear. However, the challenges faced by hypertensive adults with HIV may be different compared to the general population. PLHIV have more non-traditional CVD risk factors [3]. In addition, they may be unlikely to seek care outside of the HIV clinical setting and must often navigate perceptions of disease-related stigma [16, 17].

In order to design interventions that address challenges unique to this patient population, it is crucial to understand the experiences of PLHIV who have navigated hypertension treatment. However, other than a few studies [18–22], the barriers to hypertension awareness, treatment and control among HIV-infected adults in SSA remain largely unexplored. To better understand barriers to hypertension management among PLHIV, we conducted a qualitative study exploring the perspectives and experiences of patients with both HIV and hypertension in northern Tanzania.

## Methods

This qualitative study involved in-depth interviews with 13 individuals who were actively involved in HIV care and reported a history of hypertension. To ensure rigor and

reproducibility, the presentation of methods and results follow the Consolidated Criteria for Reporting Qualitative Research (COREQ) guidelines (S1 File) [23].

## Setting

This study was situated in the Moshi urban district of northern Tanzania and was conducted at two HIV clinics located in government-funded primary health centers, that combined, see approximately 2300 adults (1700 women and 600 men) with HIV per year. In both health centers, HIV care is typically provided by nurses and clinical officers in the HIV Care and Treatment Center. Hypertension care is managed separately from HIV care by a medical doctor or clinical officer in a different department.

## Study population

Semi-structured in-depth interviews were conducted with 13 patients with HIV and hypertension. Participants were eligible if they were active patients in one of the two HIV clinic study sites and had a self-reported diagnosis of hypertension as reflected in the criteria outlined below.

## Procedures

Eligible participants with known HIV and hypertension were recruited by the HIV clinic and research staff between September 1st and November 26th, 2018. During their routine HIV appointments patients were asked by the HIV clinic nurse or by the study nurse if they: 1) had a known diagnosis of hypertension, 2) were ever told by a health provider they had high blood pressure, and 3) had ever used medications to control blood pressure. If a patient met any of these criteria and also reported using ART for a minimum of 12 months, the study nurse invited the patient to participate in the study.

The in-depth interview guide was developed by an interdisciplinary team of physicians, nurses and social scientists from Tanzania and the United States with expertise in hypertension or HIV. The guide included open ended questions on key domains of interest, with each question followed by a list of possible probes to guide the conversation (S2 File). The guide was reviewed by Tanzanian collaborators to ensure appropriate terminology and cultural saliency. All interviews were conducted in a private room at the study site by a female physician researcher from the US with graduate level training in qualitative methodology (PM), who was not involved in clinical care at the study sites. The interviews were conducted in English and were simultaneously translated in Swahili in real-time by a female Tanzanian research assistant (LM) who had received training in qualitative methodology. Interviews lasted 59 to 157 minutes with a median duration of 105 minutes, were audio-recorded with participant consent, and were subsequently transcribed in English. A debriefing meeting was held between the interviewer (PM) and the research assistant (LM) following every interview. During the debriefing meetings preliminary themes were identified and new information added from each interview was discussed. A third member of the research team (JTH) listened to 15% of all audio recordings to confirm and clarify the on-site translation.

## Data analysis

Data analysis was conducted using applied thematic analysis [24] guided by a phenomenology theoretical framework, which is well suited to understanding individuals' lived experiences and the subjective nature of their health seeking processes [25, 26]. Transcripts were reviewed, and memos (5–19 pages each) were written for each transcript to summarize the content

emerging across the domains of interest [27]. After reviewing all memos, a codebook was developed. The transcripts were independently coded in NVivo by a physician researcher from the US (PM) to label the transcript content with the respective themes. A Tanzanian member of the research team (LW) double-coded 3 (20%) of the transcripts, and the two team members met to discuss coding protocol, reach consensus on the application of codes, and make appropriate modifications to the code book. Coding queries were used to review text coded to themes and synthesize data. Representative quotes were reviewed throughout the analysis procedure and were selected to best capture the data.

## Ethics

Study procedures were approved by Duke Health Institutional Review Board (Pro00091126), Kilimanjaro Christian Medical Centre Research Committee (No. 2265), and the Tanzania National Institute for Medical Research Ethics Coordinating Committee (NIMR/HQ/R.8a/ Vol. IX/2779). All participants were introduced to the interviewer (PM) and research assistant (LM), and were made aware of the overall purpose of the study during consent procedures. All participants provided written informed consent prior to data collection.

## Results

The characteristics of study participants are summarized in Table 1. Participants had a median age of 54 (IQR 41–65) years. No participant was diagnosed with hypertension in the HIV clinic; eight participants were diagnosed in the outpatient department (OPD), an urgent care clinic, two were diagnosed at a community health fair, one during a hospitalization, one in the preoperative setting, and one at a local pharmacy. In addition to hypertension and HIV, ten participants also reported having another chronic medical condition (including diabetes, dyslipidemia, stroke, and chronic ocular, genitourinary, gastrointestinal and cardiopulmonary issues). Of the 13 participants, eight stated they had used antihypertensive medications previously. At the time of interview, only one participant described current antihypertensive use, however, he too reported nonadherence to one of his three prescribed antihypertensive drugs.

The data revealed a range of themes across the domains of perspectives and experiences of hypertension care (see Table 2).

### Perspectives of hypertension

**Limited knowledge of hypertension.** Overall discussions revealed limited knowledge of hypertension, including what it is, its causes, progression, symptoms, complications and treatment. No respondent was able to correctly define hypertension. Most participants defined high blood pressure as what they perceived as the cause: 'thinking a lot' or 'being stressed'. A few participants also described high blood pressure as what they perceived as the symptoms: 'heart beating too fast' or the heart becoming 'tired' or 'weak'.

All participants perceived their cause for hypertension to be due to 'thinking too much' and many believed this was the sole cause of their hypertension. Thinking too much included 'being angry', 'annoyed or upset about something', 'receiving shocking news', or 'being stressed'. As explained by an interviewee:

> "When you hear shocking news, like the death of someone who is very close to you, that can lead to high blood pressure. Or if your family has problems, if they are annoying you, or disturbing you, or giving you problems that may cause high blood pressure." (Female, 45 years)

**Table 1. Characteristics of in-depth interview participants, Kilimanjaro region, 2018 (n = 13).**

| Characteristic | n |
|---|---|
| Sex | |
| Female | 11 |
| Male | 2 |
| Age | |
| 18–35 | 1 |
| 36–50 | 5 |
| 51–70 | 5 |
| 71+ | 2 |
| Marital status | |
| Single | 1 |
| Married | 2 |
| Separated/divorced | 3 |
| Widowed | 7 |
| Education | |
| None | 3 |
| Primary (grade 0–8) | 9 |
| Secondary or higher (grade 9 or higher) | 1 |
| Median duration of time since HIV diagnosis, years (IQR) | 4.0 (3.0–10.0) |
| Median duration of ART use, years (IQR) | 3.0 (2.5–6.5) |
| Median duration of time since hypertension diagnosis, years (IQR) | 1.0 (0.0–3.0) |
| Ever used antihypertensives | |
| Yes | 8 |
| No | 5 |
| Currently using antihypertensives | |
| Yes | 1 |
| No | 12 |
| Other chronic illnesses | |
| Yes | 10 |
| No | 3 |

IQR: interquartile range; ART: antiretroviral therapy.

Overall, participants believed worrying about their HIV status or being stigmatized from HIV could lead to hypertension. One participant explained:

"If you accept your HIV results and are at peace with it, you won't have high blood pressure. But if you don't accept your results, then your blood pressure can be high." (Female, 54 years)

A few respondents vocalized concern that their ART may be the cause of their hypertension. As explained by a 40-year old woman:

"I once asked the (HIV) doctor about the medication (ART). Like see, 'I have pressure. Do you think the medication (ART) I take is what's causing it?'. . .I was worried, because before I started taking the HIV medication I didn't have high blood pressure." (Female, 40 years)

Knowledge regarding unhealthy lifestyle behaviors was mixed. All but one participant identified a diet high in salt and fat as a cause for hypertension. Few participants identified other

**Table 2. Dominant themes of hypertension care among patients with HIV and hypertension, Kilimanjaro region, 2018 (n = 13).**

| Domain | Themes |
|---|---|
| Perspectives | **Limited knowledge of hypertension** |
| | Poor understanding of definition, causes, progression, symptoms, complications and treatment |
| | Belief that "thinking too much", i.e. stress, is the major cause and stress related to HIV is a cause |
| | **Perceived overlap and comparisons between hypertension and HIV** |
| | Feeling overwhelmed with having more than one medical condition |
| | Belief that HIV is chronic and controllable versus hypertension is temporary and fatal |
| | Belief that HIV requires a daily life-long medication versus hypertension is curable |
| Experiences | **Delays in hypertension diagnosis and linkage to care** |
| | Diagnosis made only after feeling ill and presenting with symptoms |
| | Delay in linkage to care following diagnosis |
| | **Challenges with provider communication and counseling** |
| | Limited counseling from providers |
| | Misconceptions often communicated to patients |
| | **Reluctance towards antihypertensive medication** |
| | Preference for stress reduction, traditional treatment and lifestyle modification |
| | **Lack of integration for hypertension and HIV care** |
| | Separation of care |
| | HIV clinic as entry point into hypertension care |
| | **Additional structural barriers to hypertension care** |
| | High cost of care |
| | Lack of access to antihypertensive medication |
| | Staff shortages and long wait times |
| | HIV associated stigma and social isolation |

unhealthy behaviors, including sedentary lifestyle, obesity, and alcohol and tobacco use as a cause.

Participants were also unaware of the range of hypertensive values and the chronicity of hypertension. Most participants assumed it was unusual to have hypertension and be asymptomatic and felt that being aware of their symptoms was sufficient enough to tell if their blood pressure was high. 'Instantaneous death', 'falling down suddenly', 'rupture of blood vessels', and 'stroke' were commonly believed hypertension complications.

All participants believed that stress reduction was the most effective way to treat hypertension. One woman explained:

"To reduce blood pressure, maybe someone should avoid (environmental) noise and try to reduce stress in their lives and conflicts. . .When there is a problem you know you can't solve, you should just try not to think about it. Like me for example, I want to build a house, but I can't buy bricks. I can't buy the cement. I can't buy the sand I need. So, I don't want to constantly think about it. I just ask God, please help me with this because in the meantime I can't handle this." (Female, 45 years)

Five participants also believed that using herbal or traditional treatment could control blood pressure, three participants believed that temporary use of antihypertensive medication could cure hypertension, and two participants believed that hypertension could be cured through prayer. Participants were unaware that antihypertensive medication use was typically life-long.

All participants reported learning about hypertension from other members of the community including hypertensive family members, friends, neighbors or co-workers. A few participants cited learning about hypertension from a medical provider.

**Perceived overlap and comparisons between hypertension and HIV.**   Patients expressed concern about having both HIV and hypertension and described having two illnesses as 'overwhelming'. They alluded to pill burden and felt that taking two medications for two diseases was 'tiresome' and 'too much'. One interviewee stated:

> "It's tiresome. It's really tiresome because I have to take antiretrovirals every day, every single day. And adding another medication is tiresome." (Female, 55 years)

While HIV was universally perceived as a chronic disease that was controllable, most participants believed that hypertension was temporary but almost always fatal. Hypertension was believed to be more dangerous than HIV because it could lead to instantaneous death, whereas HIV could be controlled with medication and one could still live a long and healthy life. One woman stated:

> "I think it's better to have HIV than pressure. . .You know with HIV you can just take medication and you will be fine. You can just continue with your life. But with pressure it can just go up suddenly and you can die instantly." (Female, 44 years)

Most participants believed that there were alternative options to antihypertensive drugs for blood pressure control, whereas they believed the only way to control HIV was through ART. A 55-year old woman stated:

> "I've never thought about stopping using my HIV medications because I know I have to use it for the rest of my life. . .But for high blood pressure? I guess maybe there will be a cure or an alternative." (Female, 55 years)

## Experiences of hypertension care

**Delays in hypertension diagnosis and linkage to care.**   Blood pressure was usually only measured when participants felt ill and presented to a health facility with symptoms, such as a headache, dizziness, fatigue or insomnia. All participants reported being diagnosed with hypertension when they presented with symptoms. One interviewee described how she only checks her blood pressure when she feels unwell:

> "I check my blood pressure once in a while because sometimes I may receive some scary news and hear things that worry me. And when things like that happen my heart starts beating fast and I get short of breath. When I feel like that I go to the dispensary and check my pressure." (Female, 45 years)

Several participants noted a delay in linkage to care following their initial hypertension diagnosis or were not linked to care at all. One participant, a 67-year old man, stated he first heard his blood pressure was high more than 3 years ago when it was measured in the preoperative setting. Although at the time, he was told his surgery was canceled due to high blood pressure, he received no counseling, linkage to care or management for his hypertension. He reported only recently becoming aware that he had hypertension when he presented to his HIV clinic with a headache and was referred to the OPD for a blood pressure check.

**Challenges with provider communication and counseling.** Respondents reported receiving minimal counseling from their healthcare providers regarding the causes, symptoms, complications and treatment of hypertension. When medications were prescribed, respondents noted they often did not receive counseling regarding side effects, potential drug-drug interactions, treatment duration or medication adherence. Participants reported that plans for blood pressure monitoring and follow-up were often not discussed.

In addition, participants described hypertension misconceptions communicated by their medical providers. Participants conveyed that healthcare providers told them that 'thinking too much' was the cause of their hypertension. Some respondents said that their providers told them if they stopped worrying about their HIV their blood pressure would improve. One woman who had a miscarriage described being told by her provider that her hypertension was due to her grief. Another woman described how a nurse implied that her hypertension was due to conflicts with her husband.

"He (the doctor) didn't discuss much about the cause but he told me that it may be because of thinking too much and being stressed out. The nurse even asked me, 'Do you have problems with your husband?' And I said, 'Yes Nurse, I've had an argument again', and she said, 'Okay, you should not think too much about these things'." (Female, 39 years)

Some participants referred to situations where a provider advised them either not to initiate antihypertensive medications or advised them to stop and 'take a rest' from their medication because their blood pressure was now in a normal range or because they were asymptomatic.

**Reluctance towards antihypertensive medication.** In general, participants appeared reluctant towards antihypertensive medications. Participants who reported previous antihypertensive use typically only used their medication when feeling symptomatic or until their prescription ran out. Reasons for discontinuing medication included high costs, lack of drug availability, drug side effects and concern for toxicity, pill burden, and lack of awareness about the silent and chronic nature of disease.

All respondents reported that they tried to 'reduce their thinking', i.e. decrease stress, in order to control their blood pressure. A few participants had tried herbal, natural and traditional treatment including drinking lots of water, eating cucumbers, using herbal plants, eating ginger and garlic supplements, and visiting a traditional healer. Participants who used traditional and herbal treatment typically stopped their antihypertensive drugs and did not discuss use of their traditional treatment with their clinical providers. One participant described her preference for herbal treatment below:

"I thought it would be too much for me. I already have this medication (for HIV), and now starting this medication (for hypertension) that I have to take for a long time? I didn't want to get used to it (antihypertensive medication), and I decided I would stick with more natural methods, and sometimes I even felt better. . . After eating garlic sometimes I would go and check my blood pressure, and I found that my pressure was a bit lower." (Female, 55 years)

**Lack of integration for hypertension and HIV care.** Participants reported that HIV and hypertension care were managed separately. Participants revealed that their HIV providers did not know they were hypertensive and that their hypertension providers did not know they had HIV. Respondents did not discuss their HIV with their hypertension provider, did not discuss their hypertension with their HIV provider, and reported that their providers did not ask them if they had other medical issues or were taking other medications. The majority noted that

their blood pressure was never measured in the HIV clinic. However, the HIV clinic was often the entry point into hypertension care. Respondents presented with symptoms to the HIV clinic, often during their regularly scheduled appointment, and were subsequently referred to the OPD for evaluation of their blood pressure. An interviewee explained:

> "No, I've never checked my blood pressure at the HIV clinic. There is not even a doctor at the HIV clinic. I've just seen nurses providing the HIV medication. . .If you tell them that you're sick they tell you to go down there (pointing to OPD)." (Female, 39 years)

Although hypertension and HIV care were separate, all respondents conveyed a preference for integrated care due to convenience and efficiency.

> "I think it will save a lot of time because I would get my (HIV) medication and then see the doctor right there (for hypertension). It would be very nice, and I think would help improve my health and help me feel a lot better." (Female, 55 years)

**Additional structural barriers to hypertension care.**   Additional structural barriers to hypertension care were described by all participants. Financial barriers were the most common. Participants reported that the high costs of hypertension care, including costs for antihypertensive medication, provider visits, transport to the clinic, and the expense of a healthy lifestyle, made it difficult to manage their hypertension. One participant stated that the cost of medication was the reason for non-adherence.

> "I'm not using (blood pressure medication). . .It's hard to raise the money now. You need money for food. You need money for this medication. If you suddenly get a fever you need money again, so how are you going to manage it?" (Female, 78 years)

Participants also voiced concern about pharmacy outages of the medication, staff shortages and long wait times. HIV stigma was an over-arching experience of participants. Many respondents reported feeling stigmatized from their HIV by their family and community and described a feeling of social isolation. A 44-year old woman described the following:

> "My brothers stigmatize me a lot. Sometimes when I want to go to bed, they insult me the whole night. Sometimes they take stones and throw them on the roof, and I feel like I am shocked and fall down. So sometimes I don't even stay at my own home." (Female, 44 years)

Stigmatization may have played a role in lower health seeking behaviors and led to HIV and hypertension nondisclosure to medical providers and subsequent disengagement in hypertension care.

## Discussion

This study explored hypertension care perspectives and experiences among hypertensive adults engaged in HIV care in northern Tanzania. To our knowledge, only two other qualitative studies have previously described hypertension care experiences among patients with HIV in SSA [18, 21]. Participants in our study described barriers unique to PLHIV including HIV-related stigma, siloed health systems, and burden from having multiple medical conditions. Participants also described barriers commonly faced by hypertensive individuals from the general population, such as limitations in knowledge, insufficient counseling, financial constraints

and challenges with access to hypertension care and treatment. While participants in our study were disengaged in hypertension care, they remained completely engaged with their HIV care. These findings suggest that the barriers depicted are unique to hypertension care, are not encountered in HIV care, and that the HIV clinic may provide a robust model of chronic disease management that can support retention in care. High quality, culturally salient, evidence-based interventions that are integrated within the HIV clinic are urgently needed to address the double burden of disease in PLHIV.

Poor and inaccurate hypertension knowledge among patients was identified as the predominant barrier to hypertension awareness, treatment and control in our study. Universally, all participants believed that "thinking too much" was the major, if not sole, cause of their hypertension, and that one could control hypertension by"reducing thinking". The concept of "thinking too much" is an idiom of anxiety, depression and distress in many settings in SSA [28]. While anxiety may lead to a relative increase in blood pressure, it is not a major contributor of uncontrolled hypertension and stress reduction is not recognized as effective primary hypertension management [29, 30]. Despite this, there was an ingrained belief in our sample that psychosocial stressors were the main risk factors for hypertension. These findings, along with poor understanding of hypertension symptoms, complications, and treatment, are consistent with data on HIV-uninfected hypertensive patients from other studies in SSA [31–33]. In contrast, participants had good knowledge about HIV disease and treatment. This may be due to concentrated efforts towards patient-directed HIV education and large-scale HIV awareness campaigns in SSA, and highlights the importance of health education initiatives in treatment adherence and clinical outcomes. Several studies in SSA have shown an improvement in hypertension outcomes following educational interventions [34, 35]. Therefore, the implementation of patient-directed hypertension educational interventions both within the community and HIV clinical setting may be one approach towards improving cardiovascular outcomes among PLHIV. In addition, participants in our study reported mainly learning about hypertension from members of the community. Given the success of peer education in HIV care in SSA [36–38] the role of peer educator in NCD management should be strongly considered and further investigated.

Stress and HIV-related stigma were overarching themes described by all participants in our study. Participants believed that stress related to living with HIV caused hypertension. They also described feeling overwhelmed and "burdened" from having more than one medical condition. Moreover, some participants described experiences of HIV-related stigma which may have led to additional stress, anxiety, depression and social isolation. Chronic untreated mental health conditions and HIV-related stigma are well-known predictors of ART nonadherence and poor HIV clinical outcomes [39–41]. In addition, stigma avoidant behaviors, i.e. those actions patients take to avoid experiencing stigma such as concealing ART or non-disclosure of HIV status, may also contribute to nonadherence and disengagement with the medical system [42]. Furthermore, evidence suggests that chronic mental health conditions such as depression may also be a predictor for hypertension treatment nonadherence in similar settings [43]. Therefore, it is possible that psychosocial stressors, disease-related stigma and stigma-avoidant behaviors contributed to hypertension care disengagement and nonadherence in our sample. Further exploration of mental health conditions, stigma and their effects on clinical outcomes in HIV and other chronic diseases among PLHIV in SSA should be prioritized to better understand the factors contributing to care disengagement. In addition, due to stigmatization, patients with HIV may feel more comfortable navigating NCD care within the HIV clinical setting with providers in which an established and trusting relationship already exists. The effects of integrating health care on HIV-related stigma and mental health outcomes among PLHIV should also be further investigated.

The participants in our study universally described high levels of HIV care engagement and ART adherence, and appeared highly knowledgeable and empowered in their HIV care. In contrast, no participant was adherent with their hypertension treatment or follow up, and hypertension knowledge was suboptimal. Therefore, the barriers described by our participants appear to pertain only to their hypertension care and were not encountered in their HIV care. In addition, the barriers described by our participants were actually quite similar to the barriers faced by the general population, however, PLHIV must also navigate HIV-related stigma and multiple siloed care systems to treat both their HIV and hypertension. Our findings are consistent with other qualitative studies exploring hypertension care barriers among PLHIV in similar settings [18, 21], and signify the critical role of the HIV clinic. Given that patients are routinely engaged in and committed to their HIV care, this setting represents an ideal opportunity to improve access of chronic disease care, beyond HIV. Our findings suggest that integrating a horizontal model of healthcare with existing vertical HIV healthcare systems may expand access to NCD care and enhance patient satisfaction and clinical outcomes [44].

HIV and hypertension were managed in discrete silos in our setting, and participants described the lack of integration as an impediment to their care. The separation of HIV and hypertension care represents a missed opportunity to improve cardiovascular outcomes among a high-risk population. Primary care and NCD models in SSA are either non-existent or are fragmented and weak. HIV is the first large chronic care program in SSA and has received significant investments. HIV clinics serve as models for robust service delivery and may provide a good opportunity to integrate NCD management. In 2011 the UN declared that HIV programs from low and middle-income countries should be leveraged for effective integration of NCDs [45, 46]. Furthermore, the Tanzanian government has recognized the increased prevalence of NCDs among PLHIV and the potential critical role of the HIV clinic. Recent national guidelines now support the integration of care for HIV and other diseases including hypertension and other NCDs [47], and an increasing number of studies suggest that the integration of HIV and NCD care in SSA is feasible, efficacious and cost effective [46, 48, 49]. However, in order to achieve successful integration of the routine assessment, prevention and management of NCDS, it will be critical to address the existing structural barriers and ensure access to trained staff, equipment, medications and protocols for NCD care within the HIV clinical setting [19, 50].

This study has some limitations. Firstly, inclusion criteria for hypertension was based on a self-reported diagnosis instead of a clinical diagnosis. Therefore, it is possible that some participants were not truly hypertensive, and thus fittingly did not engage in hypertension care. Secondly, social desirability bias when speaking with a non-Tanzanian physician may have influenced participants' responses. Thirdly, our recruitment strategy excluded patients who had dropped out of HIV care. Therefore, we may have lost some additional insight into perspectives and experiences of an HIV-disengaged population. Fourthly, we also recognize that this is a relatively small qualitative sample size [51]. Although our analysis strategy identified data saturation by participant number 13, it is possible that recruitment from a larger sample may have yielded additional emerging themes. Lastly, only two men participated in this study and therefore findings may be biased towards a female perspective. The challenge of recruiting men is not unique to our study. Men may be less likely to seek healthcare in SSA [52, 53]. Additional research to explore male perspectives of hypertension care is needed.

In conclusion, hypertensive adults engaged in HIV care in northern Tanzania described multiple, intersecting challenges to hypertension management. Multifaceted strategies that address structural barriers and seek to improve hypertension education, address psychosocial stressors and stigma, and that are integrated within HIV care, are urgently needed to improve cardiovascular outcomes among PLHIV. As more individuals with HIV initiate ART and

continue to age, the prevalence of hypertension and its associated cardiovascular morbidity and mortality among PLHIV will almost certainly continue to rise. Thus, it is imperative to develop and implement effective strategies for integrating cardiovascular care in the HIV clinical setting.

## Supporting information

**S1 File. COREQ checklist.**
(DOCX)

**S2 File. In-depth interview guide.**
(DOCX)

## Acknowledgments

We would like to thank the Duke Global Health Institute, the Duke Hubert-Yeargan Center for Global Health, and all the staff of the Pasua Health Center, Majengo Health Center, Kilimanjaro Clinical Research Institute and the KCMC-Duke Collaboration for all their efforts and support for this study. We give special thanks to Pilli Nyindo for her integral role in recruiting respondents for this study.

## Author Contributions

**Conceptualization:** Preeti Manavalan, Nathan M. Thielman, Nwora Lance Okeke, Blandina T. Mmbaga, Melissa H. Watt.

**Data curation:** Preeti Manavalan.

**Formal analysis:** Preeti Manavalan, Lisa Wanda.

**Funding acquisition:** Preeti Manavalan.

**Investigation:** Preeti Manavalan, Linda Minja.

**Methodology:** Preeti Manavalan, Linda Minja, Julian T. Hertz, Melissa H. Watt.

**Project administration:** Preeti Manavalan.

**Resources:** Preeti Manavalan, Blandina T. Mmbaga.

**Software:** Preeti Manavalan, Lisa Wanda.

**Supervision:** Preeti Manavalan, Blandina T. Mmbaga.

**Validation:** Preeti Manavalan, Linda Minja, Julian T. Hertz.

**Visualization:** Preeti Manavalan.

**Writing – original draft:** Preeti Manavalan.

**Writing – review & editing:** Preeti Manavalan, Linda Minja, Lisa Wanda, Julian T. Hertz, Nathan M. Thielman, Nwora Lance Okeke, Blandina T. Mmbaga, Melissa H. Watt.

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
