## [Decision Letter · Decision Letter 0]

15 May 2020

PONE-D-20-04882

“It’s because I think too much”: Perspectives, experiences and opportunities for improvement among adults with hypertension engaged in HIV care in northern Tanzania

PLOS ONE

Dear Dr. Manavalan,

Thank you for submitting your manuscript to PLOS ONE. After careful consideration, we feel that it has merit but does not fully meet PLOS ONE’s publication criteria as it currently stands. Therefore, we invite you to submit a revised version of the manuscript that addresses the points raised during the review process.

We would appreciate receiving your revised manuscript by Jun 29 2020 11:59PM. To enhance the reproducibility of your results, we recommend that if applicable you deposit your laboratory protocols in protocols.io, where a protocol can be assigned its own identifier (DOI) such that it can be cited independently in the future. For instructions see: http://journals.plos.org/plosone/s/submission-guidelines#loc-laboratory-protocols

We look forward to receiving your revised manuscript.

Kind regards,

Webster Mavhu

Academic Editor

PLOS ONE

Journal Requirements:

2. Please include additional information regarding the interview guide used in the study and ensure that you have provided sufficient details that others could replicate the analyses. For instance, if you developed a guide as part of this study and it is not under a copyright more restrictive than CC-BY, please include a copy, in both the original language and English, as Supporting Information.

Additional Editor Comments (if provided):

• This paper could be more nuanced. A lot of described issues relate to what is known already about hypertension knowledge, attitudes and experiences.

See for example: ‘Participants revealed multiple, intersecting challenges related to hypertension management including poor hypertension knowledge, insufficient hypertension counseling, financial constraints, lack of access to antihypertensive medications, staff shortages, HIV-related stigma, and lack of integration between hypertension and HIV care’. Just the last two themes specifically apply to people living with HIV - the rest are true for everyone else in this setting. The added value of this study is therefore not apparent.

• This paper has the potential to highlight if and how experiences of individuals with both HIV & hypertension differ from those with just hypertension or even the general population and importantly how the care of these conditions could be improved especially as there is an opportunity to manage both at once.

• Even for a qualitative study, the sample is too small. Authors state that this was guided by need to achieve theme saturation. It is unlikely that saturation was reached with just 13 in-depth interviews. Generally, 25-35 IDIs are considered acceptable for development of themes. See qualitative sampling recommendations in: Guest G, Bunce A, Johnson L. How many interviews are enough? An experiment with data saturation and variability. Field Methods. Feb 2006;18(1):59-82. Morse JM. Determining sample size. Qualitative Health Research. Jan 2000;10(1):3-5.

• Also, if sampling was guided by the need to achieve theme saturation, how did authors determine they had reached theme saturation? Often, this is achieved by collecting initial interviews, reviewing emerging issues, collecting additional ones, reviewing again until no new themes emerge. As already stated, this process is rarely achieved after just 13 interviews. If researchers conducted 13 interviews due to pragmatic and other considerations, this should be stated. Otherwise, the data collection and analysis process leading to theme saturation should be fully described.

• Themes could be more analytical than descriptive. For example ‘Poor understanding of the causes of hypertension’ could be a sub-theme of ‘Poor hypertension knowledge’ i.e. causes, mitigation etc.

• The way the themes /sub-themes are presented is confusing. For example, ‘lifestyle factors’ is listed as a sub-theme of ‘Poor understanding of the causes of hypertension’. Should it be ‘lack of knowledge of lifestyle factors’? Even so, there is some contradiction on this theme as initially, it appears participants were unaware but lines 326-27 they mention some of the lifestyle measures they took to manage hypertension – suggesting they know the lifestyle factors. This is where a more analytical reflection would help.

• Separately, “thinking too much” is an idiom of anxiety and depression in many African settings – it is possible this can lead to hypertension?

• Still on themes, authors say they identified 10 themes and 11 sub-themes. Table 2 presents dominant themes – lists domains, 8 themes and 23 sub-themes?

• Interview process is unclear. Did English-speaker interview together with Swahili RA with the RA interpreting every question and response? If so, probably explains length of interviews?

• Setting is given as 1 of 2 clinics (abstract) and also 2 clinics (line 67).

• Abstract should be structured as follows: Background, Methods, Results, Conclusions

• Please attach interview guide

Reviewers' comments:

Reviewer's Responses to Questions

**Comments to the Author**

1. Is the manuscript technically sound, and do the data support the conclusions?

Reviewer #1: Yes

Reviewer #2: Yes

2. Has the statistical analysis been performed appropriately and rigorously? 

Reviewer #1: N/A

Reviewer #2: Yes

3. Have the authors made all data underlying the findings in their manuscript fully available?

Reviewer #1: No

Reviewer #2: No

4. Is the manuscript presented in an intelligible fashion and written in standard English?

Reviewer #1: Yes

Reviewer #2: Yes

5. Review Comments to the Author

Reviewer #1: This paper demonstrates well the huge differences invested in treatment literacy for HIV opposed to hypertension, a gap that needs urgently addressing. As HIV cohorts age addressing these comorbidities is increasingly important and urgently need to invest in improving care.

Overall the paper is well written and the thematics raised cover the important perceptions and barriers to accessing hypertension care for people living with HIV

It’s because I think too much”: Perspectives, experiences

and opportunities for improvement among adults with

hypertension engaged in HIV care in northern Tanzania

Would suggest edit to the title: Perspectives, experiences and opportunities for improvement in care of adults living with HIV and hypertension in northern Tanzania

Lines 36-43 – in the introduction it would be useful to also include some reference to the prevalence and response if known in Tanzania to add to the broader statistics quoted.

Line 58 – consider referencing these other studies looking at perceptions of hypertension in HIV cohort in SSA including from Tanzania – maybe highlight some key findings from these papers and can be developed further in discussion how authors findings compare

Weiss JJ, Konstantinidis I, Boueilh A, et al. Illness Perceptions, Medication Beliefs, and Adherence to Antiretrovirals and Medications for Comorbidities in Adults With HIV Infection and Hypertension or Chronic Kidney Disease. J Acquir Immune Defic Syndr. 2016;73(4):403–410. doi:10.1097/QAI.0000000000001075

Hing M, Hoffman RM, Seleman J, Chibwana F, Kahn D, Moucheraud C. 'Blood pressure can kill you tomorrow, but HIV gives you time': illness perceptions and treatment experiences among Malawian individuals living with HIV and hypertension. Health Policy Plan. 2019;34(Supplement_2):ii36–ii44. doi:10.1093/heapol/czz112

Kagaruki GB, Mayige MT, Ngadaya ES, et al. Knowledge and perception on type2 diabetes and hypertension among HIV clients utilizing care and treatment services: a cross sectional study from Mbeya and Dar es Salaam regions in Tanzania. BMC Public Health. 2018;18(1):928. Published 2018 Jul 28. doi:10.1186/s12889-018-5639-7

Line 68 needs editing – maybe better to state number of patients in care at each facility. Some more detail on site choice would be helpful – were these tertiary , district or primary care facilities for example. Do the authors know in the chosen facilities who ( doctor or nurse) provides HIV and BP care. Is care integrated in the selected sites or we know up front that there are different clinics?

Line 72: would suggest this reflects the criteria outlined in lines 77-79 re who eligible for recruitment

Line 117 – don’t need to give range of age – just state median with the IQR

Line 119 – median time HIV disease – do you mean the time from HIV positive diagnosis? Bit worrying such a big gap between the two

Line 121- did the authors pick up where the hypertension diagnosis had been made if after the HIV diagnosis – in ART clinic- at another primary care clinic?

Table 2: Listing lifestyle factors under poor understanding needs some explanation – were they not aware of diet , exercise etc

Line 370: edit queue

Line 373 : may be worth putting USD equivalent

Line 417 – agree the importance of health education but would also add on the importance placed on engaging people living with HIV in their care and the role of peer support to strengthen treatment literacy – this point I feel is also missing within the conclusion as a key message

Line 459/60 – the authors may also like to reference the recommendation on screening for cardiovascular disease within the WHO 2016 guidelines for antiretroviral care

Do the authors know if there is a specific recommendation within the latest Tanzanian national guidelines for integration of ART and hypertension care - is the principle at least supported in national policy?

In the discussion on integration and service delivery the authors may also like to consider adding whether using the principles of differentiated service delivery- which Tanzania has adopted for HIV within their operational manual – to enhance hypertension care. I believe there is guidance within that document recommending NCD /HIV integration, so to highlight if policies are supported versusu the challenges to fund and implement them

Reviewer #2: This is a very well written, clear manuscript which discusses the experiences of people living with HIV and hypertension in Tanzania, and reflects upon the challenges of managing this ‘dual burden’ of disease. Integration is an important issue to consider, especially in contexts with high HIV/NCD prevalence and where there are significant differences in the provision of medication (eg ARVs being free but hypertensive treatment not).

There are limitations to the paper which are acknowledged by the authors, and which mostly relate to the study design. As participants self-reported their diagnosis of hypertension, there may have been recall bias or uncertainty around this. There is also a very small sample size – this is a qualitative study, so a small sample is expected, but the authors rely on data from only 13 patients making me question how thorough the analysis could be.

The perspectives of health-care providers are also missing, which would have been valuable in terms of making practical recommendations about how outcomes could be improved and understanding challenges from their perspective.

Despite the limitations of the study design, the paper is well structured, the data well-presented and the discussion reflects upon many important, relevant issues to contexts within SSA, not just Tanzania.

Title

The manuscript focussed mostly on experiences and perspectives and it is unclear what the ‘opportunities for improvement’ are eg clinical outcomes, cardiovascular outcomes, HIV outcomes, improvement in psycho-social well being…? I suggest removing ‘opportunities for improvement’ from the title. The rest of the manuscript refers to perspectives and and experiences throughout, so I would edit the title to be in line with this.

Abstract

Overall, the abstract is well written. I would like to see some additional figures about HIV and hypertension prevalence in Tanzania if the word limit allows.

Introduction

Line 52 – write out twofold in full

Line 53 – persons with HIV could be written in the standard PLWH which is used throughout

Materials and methods

The methods section is very clear and detailed. COREQ guidelines are not mentioned – was the checklist adhered to and submitted, as per journal requirements?

Line 63 – as this is a qualitative paper, in line with other papers in the journal I suggest removing ‘materials’ as these are not relevant to this study

Line 68 – why were these two clinics chosen? The abstract makes it sound like the study was conducted in ‘one of two’ clinics so I am unsure if it was one clinic, or both. Throughout the results, the clinics of the patients are not mentioned or linked with the quotes. Were there any differences between the two sites, or were they grouped together throughout rather than analysed separately?

Line 73 – rather than recruitment being conducted until thematic saturation was reached, this should read ‘data collection was carried out until thematic saturation was reached’.

Line 76 – who specifically were the HIV clinic and research staff? Was more than one person doing the recruitment at each site? A research assistant is mentioned later on, but it is unclear of how many different people were involved in the recruitment and data collection, and what their precise roles were.

Line 78 – as mentioned by the authors, self-reported hypertension status is a limitation to the study. Would it have been possible to verify this through patient records?

Line 90 – the PI is referred to as a physician throughout, without mentioning their role as a researcher, or research training (in line with COREQ guidelines). Was the physician also a qualitative researcher? There are also other details missing from COREQ here (eg gender of interviewer/assistant).

Line 93 - I am very surprised that the interviews took so long as most of my interviews on similar issues do not take more than 30-45 minutes, so I would like to get some tips! Two hours is a very long time, especially when conducting interviews with people who – as the authors state – were often unsure of the treatment that they had received. Can the authors attach a copy of the interview guide in order to see the questions that were asked? As per COREQ guidelines, can you also state the mean duration and be more precise with interview length eg between 50 and 97 minutes.

Line 94 – 3rd should be written in full

Line 95 – at what point were the audio-recordings translated? Was the transcription and translation carried out simultaneously by the person transcribing?

Line 105 and prior - as there is a such a small number of interviews, I would give the number that were verified rather than the percentage (or both)

Line 114 - which language was the consent given in? Can the authors reflect upon the literacy levels in this population to explain why written rather than verbal consent was taken? This line should probably be rephrased to read ‘prior to data collection’ rather than ‘prior to enrolment’.

Results

Line 118 – I suggest rephrasing ‘no more than a primary school education’ – I understand what the authors mean here, but the phrasing is a little judgemental.

Line 119 - would ‘median duration of HIV disease’ be ‘time since HIV diagnosis’ (see also table)? Is this information self-reported or from clinical records? (As an aside, if clinical records were consulted, was this mentioned in the consent, and who accessed them?)

Line 124 – 8 should read eight

Table two, line 135 onwards – are some of the phrases under the themes ‘in vivo’ codes from NVivo, written exactly as the interviewees spoke them? It would be helpful to point these out, especially as some of them look grammatically incorrect in the context of a table.

I did not understand what ‘diagnosis in setting of symptoms’ meant -can you explain? It may be a clinical term I am unfamiliar with.

Medications should be singular, not plural

Overall, the data presented in the results in the form of quotes is well-written but I have some general comments on the presentation and introduction of the data. The quotes need more introduction eg ‘As explained by one of the interviewees below:’ or ‘As a 28 year old female interviewee stated:’ rather than just being listed. Not all the quotes link directly to the points being made prior to their introduction so some restructuring is required.

There are also some spacing and formatting issues which make the quotes difficult to read, and as PLoS One does not do copy editing, it would be helpful to check the formatting and allow more space between the quotes and the body of the text.

As there are a very small number of interviewees, I would prefer the authors to be more specific if they can. Whilst the aim of a qualitative paper is not to quantify responses (and I don’t want to suggest a full quantification of quotes at all), I feel in this case it would be beneficial to specify when it was only one participant, or all participants except for one etc rather than an over-reliance on many/few/some.

Line 163 - again, I understand what the authors mean here but rather than stating ‘poor levels’, is there a less judgemental way of phrasing this? Limited understanding, or lower levels, for example. I would also question whether calling high blood pressure ‘presha’ implies a poor level of knowledge, as suggested in the previous sentence, or is simply a very common Sub-Saharan African way of referring to it and thus implies an understanding of the condition.

Line 177 – this is interesting – can you provide more examples of those who talked about instantly dying or falling down?

Line 184 - this is a very long sentence, and the interesting story gets lost! Can you revise it?

Line 189 - should this be pharmacological treatment?

Line 194 – how was ‘noise’ translated? Can you refer back to the transcript? I wonder if this is about literal noise, or about avoiding stress or chaotic situations rather than noisy environments… This is an interesting point which I would like to understand more.

Line 181 – how many were many? I am interested to know how many used religious or traditional treatments.

Line 151 – the quote does not link back to the previous statement about side-effects

Line 159 – should this be a few/some participants, rather than ‘few’?

Line 204 – how was this question asked? If it was a closed ended question, I could imagine most people would say yes to wanting additional information, but I am interested to know if people asked for extra information themselves, without prompting.

Line 207 – my concern with this section is that not everyone was on treatment, thus there is a smaller sample interviewed here. Can the authors clarify at this point how many people were on treatment, and how big the sample is in reference to pill burden. Similarly, on line 212, how many participants from the sample were asked this question, as presumably it wasn’t asked to those who were on treatment (which I think is only one interviewee).

Line 223 – this sentence is unclear to me. Do you mean challenges to adherence or interruption of treatment?

Line 226 – the word committed is somewhat problematic. Someone could be committed to taking their HIV treatment, but still be unable to do so because of financial means, stigma, unable to get to the clinic etc. The commitment is still there, but they are unable to carry out the action. The following quote does not link to the point about being ‘committed’ so I suggest the authors re-order this section.

Line 236 – the ‘setting of symptoms’ phrasing is unclear to me.

Line 245 – including the age of this person would be helpful here – it is in the quote below, but could be moved up. The quote doesn’t mention the time period of several decades ago – can you include this to give more context?

Line 249 – OPD can be left in full as it is only used once in the manuscript

Line 257 – could this be rephrased as ‘challenges with provider communication and counselling’ ?

Line 262 – cited rather than ‘did cite’

Line 264 – did this mean that they weren’t discussed regularly at follow up appointments, or that not all interviewees talked about monitoring?

Line 272 – the below quote refers to arguments, rather than conflicts – did the longer audio-recording talk about more general conflict?

Line 290 – rather than admitting reluctance, could this be rephrased to ‘were reluctant to use…’?

Line 293 – the point about only being giving a prescription for 5-7 days is interesting, but could be more about practical reasons for not adhering to treatment (eg returning to the clinic weekly) than reluctance towards taking medication. Can the authors reflect on this?

Line 300 – I appreciate that this is a person’s words, but it would be easier to read if the grammar were corrected eg the doctor told me, rather than have told me. As long as none of the meaning is lost, a ‘tidying up’ of quotes throughout the manuscript would help the readers.

Line 308 – is the reference to ‘reduce their thinking’ a direct quote from someone? Is this referring to stress or something else?

Line 315 – is traditional medicine man the term used locally, or does the more commonly used ‘traditional healer’ apply here?

Line 327 – which sub- category of ‘most’ are being discussed here? How many of the sample were not already in care – is this the same as the one person currently on treatment? How was this question phrased (eg open or closed question, or participants offered this information without prompting)?

Line 330 – this is a very programmatic turn of phrase which feels out of place in the results! (Although I completely agree with the point). ‘Participants reported that HIV and hypertensive care were managed separately…’ could be one possible rephrasing.

Line 343 – can you provide more examples here? This is only one patient, but you state that many stated a preference…

Line 374 -this paragraph feels more like an analysis/reflection for the discussion, rather than part of the results. Can you rewrite it, or move to the discussion?

Discussion

This is very nicely written and the results, literature and reflections are woven together well.

The limitations are very fair and well-reflected upon by the authors.

6. PLOS authors have the option to publish the peer review history of their article (what does this mean?). If published, this will include your full peer review and any attached files.

Reviewer #1: No

Reviewer #2: No

---

## [Author Response · Author response to Decision Letter 0]

9 Jul 2020

June 29, 2020

The Editor

PLOS ONE

Re: PONE-D-20-04882, “It’s because I think too much”: Perspectives, experiences and opportunities for improvement among adults with hypertension engaged in HIV care in northern Tanzania

Dear Dr. Webster Mavhu,

Thank you for the opportunity to provide a revision for this manuscript. We have addressed each comment and suggestion. The specific nature of the response to each point is outlined with our submission and changes are made, where appropriate, in the revised manuscript text. In particular, we have added more nuance to the manuscript by revising the discussion to highlight the key finding that the barriers to hypertension care faced by people living with HIV are specific to hypertension and not seen in HIV. Our findings suggest that the HIV clinic may support integrated health services and retention in hypertension care, in line with the national policies and recommendations. 

Due to the small sample size and the detailed content of the narrative interview transcripts which include sensitive and identifying information, it is not possible to deidentify the qualitative data. The ethics committee has not approved public release of this type of data and the data supporting the findings of this study will be available upon reasonable request. Data requests can be sent to the corresponding author or to the Duke University Health System IRB with protocol number Pro00091126, Mailing Address: 2424 Erwin Road, Suite 405, Durham, NC, Tel: (919) 668-5111, Email: eIRB@mc.duke.edu. This manuscript has not been previously published nor is it under consideration for publication elsewhere. The authors have seen and approved the version of the manuscript enclosed and concur with its submission to PLOS ONE. 

Thank you very much for considering this revision. 

Sincerely,

Preeti Manavalan, MD, MSc-GH

Infectious Diseases Fellow

Division of Infectious Disease

Duke University Medical Center

Academic Editor 

1. Comment: Please ensure that your manuscript meets PLOS ONE's style requirements, including those for file naming. The PLOS ONE style templates can be found at https://journals.plos.org/plosone/s/file?id=wjVg/PLOSOne_formatting_sample_main_body.pdf and https://journals.plos.org/plosone/s/file?id=ba62/PLOSOne_formatting_sample_title_authors_affiliations.pdf

Response: We thank the editor for this comment. We have reviewed the PLOS ONE style templates and believe that the manuscript now meets PLOS ONE’s style requirements including those for file naming. 

2. Comment: Please include additional information regarding the interview guide used in the study and ensure that you have provided sufficient details that others could replicate the analyses. For instance, if you developed a guide as part of this study and it is not under a copyright more restrictive than CC-BY, please include a copy, in both the original language and English, as Supporting Information.

Response: We thank the editor for this thoughtful feedback. We have provided a copy of the interview guide used in the study as a supporting document (S2 File. In-depth Interview Guide) and have referred to the supporting document in the methods section of the manuscript. 

3. Comment: We note that you have indicated that data from this study are available upon request. PLOS only allows data to be available upon request if there are legal or ethical restrictions on sharing data publicly. For information on unacceptable data access restrictions, please see http://journals.plos.org/plosone/s/data-availability#loc-unacceptable-data-access-restrictions.

Response: We thank the editor for this feedback. Due to the small sample size and the detailed content of the interview transcripts, including sensitive and potentially identifying information, it is not possible to deidentify the qualitative data. The ethics committee has not approved public release of this type of data, and the data supporting the findings of this study will be available upon reasonable request. Data requests can be sent to the corresponding author or to the Duke University Health System IRB with protocol number Pro00091126, Mailing Address: 2424 Erwin Road, Suite 405, Durham, NC, Tel: (919) 668-5111, Email: eIRB@mc.duke.edu. 

4. Comment: This paper could be more nuanced. A lot of described issues relate to what is known already about hypertension knowledge, attitudes and experiences.

See for example: ‘Participants revealed multiple, intersecting challenges related to hypertension management including poor hypertension knowledge, insufficient hypertension counseling, financial constraints, lack of access to antihypertensive medications, staff shortages, HIV-related stigma, and lack of integration between hypertension and HIV care’. Just the last two themes specifically apply to people living with HIV - the rest are true for everyone else in this setting. The added value of this study is therefore not apparent.

Response: We thank the editor for this thoughtful feedback. To our knowledge, there have been very few qualitative manuscripts published in the literature exploring hypertension care experiences among patients with HIV in sub-Saharan Africa (Temu TM, Bahiru E, Bukachi F, et al. Lay beliefs about hypertension among HIV-infected adults in Kenya. Open Heart 2017;4:e000570; Hing M, Hoffman RM, Seleman J, et al. ‘Blood pressure can kill you tomorrow, but HIV gives you time’: illness perceptions and treatment experiences among Malawian individuals living with HIV and hypertension. Health Policy Plan 2019;34:ii36–44). The findings from our study reveal that barriers to hypertension care faced by people living with HIV are specific to hypertension, and are actually similar to the barriers faced by the general population. Given that our population was highly engaged in HIV care, we conclude that the HIV clinic can be mobilized to support retention in hypertension care. We believe that these are important findings that contribute to the literature to support integrated health services, and we have added an additional paragraph to the discussion to highlight the significance of the key findings. In doing so, we hope to make the added value of this study more apparent and bring more nuance to the paper. 

“The participants in our study universally described high levels of HIV care engagement and ART adherence, and appeared highly knowledgeable and empowered in their HIV care. In contrast, no participant was adherent with their hypertension treatment or hypertension follow up, and hypertension knowledge was suboptimal. The barriers to hypertension care faced by PLWH appear to be specific to hypertension, since these same barriers were not encountered in their HIV care. Of note, the barriers to hypertension care faced by PLWH were actually quite similar to the barriers faced by the general population, however, PLWH must also navigate HIV-related stigma and multiple siloed care systems to treat both their HIV and hypertension. Our findings are consistent with other qualitative studies exploring hypertension care barriers among PLWH in similar settings [19,22], and signify the critical role of the HIV clinic. Given that patients are routinely engaged in and committed to their HIV care, this setting represents an opportunity to improve access of chronic disease care, beyond HIV, among PLWH in SSA. Our findings suggest that integrating a horizontal model of healthcare with existing vertical HIV healthcare systems may expand access to NCD care and enhance patient satisfaction and clinical outcomes [47].” (page 27, paragraph 2)

5. Comment: This paper has the potential to highlight if and how experiences of individuals with both HIV & hypertension differ from those with just hypertension or even the general population and importantly how the care of these conditions could be improved especially as there is an opportunity to manage both at once.

Response: We thank the editor for this comment. We agree that comparison to the general population would be helpful, and thus, have described the lack of a non-HIV comparator group as a limitation to our study and have added the following statement to the discussion: “Fourthly, our recruitment strategy excluded patients without HIV; therefore, we were unable to compare how experiences of individuals with HIV and hypertension in our setting differ from those with hypertension alone.” (page 28, lines 610-613) 

In addition, we have elaborated on the importance of integrated care in the discussion as follows: “HIV and hypertension were managed in discrete silos in our setting, and our participants described the lack of integration as an impediment to their care. The separation of HIV and hypertension care represents a missed opportunity to improve cardiovascular outcomes among a high-risk population. Primary care and NCD models in SSA are either non-existent or are fragmented and weak. HIV is the first large chronic care program in SSA and has received significant investments. HIV clinics serve as models for robust service delivery and may provide a good opportunity to integrate NCD management. In 2011 the UN declared that HIV programs from low and middle-income countries should be leveraged for effective integration of NCDs [48,49]. Furthermore, the Tanzanian government has recognized the increased prevalence of NCDs among PLWH and the potential critical role of the HIV clinic. Recent national guidelines in Tanzania now support the integration of care for HIV and other diseases including hypertension and other NCDs [50], and an increasing number of studies in SSA suggest that integrating the care of HIV and NCDs is feasible, efficacious and cost effective [49,51,52]. However, in order to achieve successful integration of the routine assessment, prevention and management of NCDS, including hypertension, it will be critical to address the system-level challenges and ensure access to trained staff, equipment, medications and protocols for NCD care within the HIV clinical setting [20,53].” (page 28, paragraph 1)

6. Comment: Even for a qualitative study, the sample is too small. Authors state that this was guided by need to achieve theme saturation. It is unlikely that saturation was reached with just 13 in-depth interviews. Generally, 25-35 IDIs are considered acceptable for development of themes. See qualitative sampling recommendations in: Guest G, Bunce A, Johnson L. How many interviews are enough? An experiment with data saturation and variability. Field Methods. Feb 2006;18(1):59-82. Morse JM. Determining sample size. Qualitative Health Research. Jan 2000;10(1):3-5.

Response: We thank the editor for this thoughtful comment. We acknowledge that the small qualitative sample size is a limitation in our study (Guest G, Bunce A, Johnson L. How many interviews are enough? An experiment with data saturation and variability. Field Methods. Feb 2006;18(1):59-82). While our ongoing analysis identified data saturation by participant number 13, it is possible that new themes may have emerged if recruitment occurred from a broader sample in the population, and this has been added as a limitation and described in the discussion as follows: “We also recognize that this is a relatively small qualitative sample size [54]. Although our analysis strategy identified data saturation by participant number 13, it is possible that recruitment from a larger sample may have yielded additional emerging themes.” (page 29, lines 613-615)

7. Comment: Also, if sampling was guided by the need to achieve theme saturation, how did authors determine they had reached theme saturation? Often, this is achieved by collecting initial interviews, reviewing emerging issues, collecting additional ones, reviewing again until no new themes emerge. As already stated, this process is rarely achieved after just 13 interviews. If researchers conducted 13 interviews due to pragmatic and other considerations, this should be stated. Otherwise, the data collection and analysis process leading to theme saturation should be fully described.

Response: We thank the editor for this comment. A debriefing meeting with the interviewer and translator, who also served as a co-interviewer, was conducted following every interview. During the debriefing meetings, preliminary themes were identified and new information and possible themes added from each interview was discussed. The methods section has been revised to reflect these procedures as follows: “A debriefing meeting was held between the interviewer (PM) and the research assistant translating the interviews (LM) following every interview. During the debriefing meetings preliminary themes were identified and new information added from each interview was discussed.” (page 7, lines 111-114)

8 & 9. Comment: Themes could be more analytical than descriptive. For example, ‘Poor understanding of the causes of hypertension’ could be a sub-theme of ‘Poor hypertension knowledge’ i.e. causes, mitigation etc.

The way the themes /sub-themes are presented is confusing. For example, ‘lifestyle factors’ is listed as a sub-theme of ‘Poor understanding of the causes of hypertension’. Should it be ‘lack of knowledge of lifestyle factors’? Even so, there is some contradiction on this theme as initially, it appears participants were unaware but lines 326-27 they mention some of the lifestyle measures they took to manage hypertension – suggesting they know the lifestyle factors. This is where a more analytical reflection would help.

Response: We thank the editor for these thoughtful comments. We have made specific changes in Table 2 and in the results section to better communicate the themes and subthemes and to clarify any previous confusion. For example, we have revised the subtheme ‘lifestyle factors’ to ‘lack of knowledge of some lifestyle factors such as sedentary lifestyle, obesity, smoking, and alcohol use as a cause’ (page 10, Table 2). We have also added the following paragraph in the text regarding knowledge of lifestyle factors: “All but 1 participant identified an unhealthy diet high in salt, fat and oil as a cause for hypertension. Few participants identified sedentary lifestyle and obesity as a cause. Only 1 participant identified alcohol use as a hypertension risk factor and no participant reported tobacco use as a cause. Chemicals and pesticides used in farming were also believed to be potential causes of hypertension.” (page 12, paragraph 2) 

In addition, we removed the subthemes ‘Limited provider counseling and communication’, ‘Limited awareness’ and ‘Separation of HIV and hypertension care’ from the theme ‘Multiple barriers to hypertension care’ in Table 2 as we felt these subthemes were already discussed in detail within the themes ‘Limited knowledge of hypertension’, ‘Challenges with provider communication and counseling’, and ‘Lack of integration for hypertension and HIV care’. (page 10, Table 2)

We fully appreciate the editor’s comment regarding analytical reflection. As a team, we decided that detailed, descriptive themes and subthemes best reflected the data and conveyed it most clearly. 

10. Comment: Separately, “thinking too much” is an idiom of anxiety and depression in many African settings – it is possible this can lead to hypertension?

Response: We thank the editor for this thoughtful comment. We agree with the editor that the concept of ”thinking too much” is an idiom of anxiety, depression, and distress in many African settings (Kaiser BN, Haroz EE, Kohrt BA, Bolton PA, Bass JK, Hinton DE. “Thinking too much”: A systematic review of a common idiom of distress. Soc Sci Med. Dec 2015; 147:170-183). While anxiety may lead to elevated blood pressure to a certain degree, it is not a major contributor to hypertension. Thus, we believed there was a misconception among the participants in our study that emotional and mental distress was a major contributor to hypertension risk and have revised the discussion accordingly to reflect upon these findings. To clarify the meaning of “thinking too much” and the potential role of stress in hypertension, we have added the following to the discussion section:

“Universally, all participants believed that ‘thinking too much’ was the major, if not sole, cause of their hypertension, and that one could control hypertension by ‘reducing thinking’. The concept of ‘thinking too much’ is an idiom of anxiety, depression and distress in many settings in SSA [30]. While anxiety may lead to a relative increase in blood pressure, it is not a major contributor of uncontrolled hypertension and stress reduction is not recognized as effective primary hypertension management. [31,32]. Despite this, there was an ingrained belief in our sample that psychosocial challenges were a major contributor to hypertension risk and by reducing these psychosocial stressors one may control blood pressure.” (page 25, paragraph 2)

11. Comment: Still on themes, authors say they identified 10 themes and 11 sub-themes. Table 2 presents dominant themes – lists domains, 8 themes and 23 sub-themes?

Response: We thank the editor for this comment. We identified 10 parent codes and 11 child codes which were then used to label the data, and the methods section has been revised to reflect this. Data analysis, including further coding analysis, then yielded 2 domains with 8 emerging themes and 20 sub-themes which are represented in Table 2. (page 10, Table 2)

12. Comment: Interview process is unclear. Did English-speaker interview together with Swahili RA with the RA interpreting every question and response? If so, probably explains length of interviews?

Response: We thank the editor for this comment. The English-speaker did interview together with the Swahili-speaker and every question and response was translated in real-time which accounts for the length of interviews. The methods section has been revised to communicate these procedures more clearly. “All interviews were conducted in a private room at the study site by a female physician researcher from the US with graduate level training in qualitative methodology (PM), who was not involved in any clinical care at the study clinic sites. The interviews were conducted in English and were simultaneously translated in Swahili in real-time by a female Tanzanian research assistant (LM) who had received training in qualitative methodology. Interviews lasted 59 to 157 minutes with a median duration of 105 minutes, were audio recorded with participant consent, and were subsequently transcribed in English.” (page 7, paragraph 1)

13. Comment: Setting is given as 1 of 2 clinics (abstract) and also 2 clinics (line 67).

Response: We thank the editor for this comment. Participants were recruited from 2 clinics and for clarification the statement has been revised to ‘2 HIV clinics’ in the abstract. 

14. Comment: Abstract should be structured as follows: Background, Methods, Results, Conclusions

Response: We thank the editor for this comment. The abstract is now structured with a Background, Methods, Results and Conclusions section.

15. Comment: Please attach interview guide

Response: We thank the editor for this comment. The interview guide has been included as a supporting document (S2 File. In-depth Interview Guide). 

Reviewer #1: 

1. Comment: This paper demonstrates well the huge differences invested in treatment literacy for HIV opposed to hypertension, a gap that needs urgently addressing. As HIV cohorts age addressing these comorbidities is increasingly important and urgently need to invest in improving care.

Overall the paper is well written and the thematics raised cover the important perceptions and barriers to accessing hypertension care for people living with HIV

Response: We thank the reviewer for this positive feedback and have addressed specific comments and critical feedback below. 

2. Comment: It’s because I think too much”: Perspectives, experiences and opportunities for improvement among adults with hypertension engaged in HIV care in northern Tanzania

Would suggest edit to the title: Perspectives, experiences and opportunities for improvement in care of adults living with HIV and hypertension in northern Tanzania

Response: We thank the reviewer for this comment. We have carefully considered the comments from reviewer 1 and from reviewer 2 regarding the proposed title change to this paper. We have changed the title to “It’s because I think too much”: Perspectives and experiences of adults with hypertension engaged in HIV care in northern Tanzania, which is more consistent with the comments from reviewer 2 as we agree that opportunities for improvement was not a major finding or focus in our study. 

3. Comment: Lines 36-43 – in the introduction it would be useful to also include some reference to the prevalence and response if known in Tanzania to add to the broader statistics quoted.

Response: We thank the reviewer for this comment. We have added a few references regarding hypertension prevalence among the general population (Geldsetzer P, Manne-Goehler J, Marcus M, et al. The state of hypertension care in 44 low-income and middle-income countries: a cross-sectional study of nationally representative individual-level data from 1.1 million adults. Lancet. 2019;394:652-662.) and among people living with HIV in Tanzania (Peck RN, Shedafa R, Kalluvya S, et al. Hypertension, kidney Disease, HIV and antiretroviral therapy among Tanzanian adults: A cross-sectional study. BMC Med. 2014;12:125; Manavalan P, Madut DB, Hertz JT, et al. Hypertension burden and challenges across the hypertension treatment cascade among adults enrolled in HIV care in northern Tanzania. J Clin Hypertens. 2020, Forthcoming). To add to the broader statistics and to provide more context to our study, the following paragraph has been revised in the introduction:

“SSA has the highest prevalence of hypertension in the world [9], and in Tanzania approximately 1 in 3 people may be hypertensive [10]. Despite this increased burden of disease, most patients with hypertension in SSA are not on treatment. A meta-analysis examining hypertension in SSA revealed that 73% of hypertensive adults were unaware of their diagnosis, 82% with a known diagnosis were not on treatment, and 93% of those on treatment had an uncontrolled blood pressure [11]. These concerning trends are also present among persons with HIV [12]: a recent study in northern Tanzania found that among a cohort of hypertensive patients enrolled in HIV care, 65% were unaware of their diagnosis, 90% were not on antihypertensive treatment and all had uncontrolled blood pressure [13].” (page 4, paragraph 2)

4. Comment: Line 58 – consider referencing these other studies looking at perceptions of hypertension in HIV cohort in SSA including from Tanzania – maybe highlight some key findings from these papers and can be developed further in discussion how authors findings compare

Weiss JJ, Konstantinidis I, Boueilh A, et al. Illness Perceptions, Medication Beliefs, and Adherence to Antiretrovirals and Medications for Comorbidities in Adults With HIV Infection and Hypertension or Chronic Kidney Disease. J Acquir Immune Defic Syndr. 2016;73(4):403–410. doi:10.1097/QAI.0000000000001075

Hing M, Hoffman RM, Seleman J, Chibwana F, Kahn D, Moucheraud C. 'Blood pressure can kill you tomorrow, but HIV gives you time': illness perceptions and treatment experiences among Malawian individuals living with HIV and hypertension. Health Policy Plan. 2019;34(Supplement_2):ii36–ii44. doi:10.1093/heapol/czz112

Kagaruki GB, Mayige MT, Ngadaya ES, et al. Knowledge and perception on type2 diabetes and hypertension among HIV clients utilizing care and treatment services: a cross sectional study from Mbeya and Dar es Salaam regions in Tanzania. BMC Public Health. 2018;18(1):928. Published 2018 Jul 28. doi:10.1186/s12889-018-5639-7

Response: We thank the reviewer for this thoughtful feedback, and have referenced these additional studies in the introduction to provide a more detailed context and background for our study. “However, other than a few studies [19–23], the barriers to hypertension awareness, treatment and control among HIV-infected adults in SSA remain largely unexplored.” (page 5, line 61)

5. Comment: Line 68 needs editing – maybe better to state number of patients in care at each facility. Some more detail on site choice would be helpful – were these tertiary, district or primary care facilities for example. Do the authors know in the chosen facilities who ( doctor or nurse) provides HIV and BP care. Is care integrated in the selected sites or we know up front that there are different clinics?

Response: We thank the reviewer for this comment and have provided additional details in the methods section regarding the two study sites, including the type of facility and number of patients seen. In addition, we have described in further detail the type of healthcare workers providing HIV and hypertension care at each study site. We have added the following paragraph to the methods section: “This study was situated in the Moshi urban district of northern Tanzania. In northern Tanzania, the approximate community prevalence of hypertension among adults is 28% [25]. The study was conducted at two HIV clinics located in government-funded primary health centers, Majengo Health Center and Pasua Health Center. Majengo Health Center sees approximately 1200 adults (900 women and 300 men) with HIV per year and Pasua Health Center sees approximately 1100 adults (800 women and 300 men) with HIV per year. In both health centers, HIV care is typically provided by nurses and clinical officers in the HIV Care and Treatment Center. Hypertension care is managed separately from HIV care, and is managed by a medical doctor or clinical officer in a different department. (page 5, paragraph 4)

6. Comment: Line 72: would suggest this reflects the criteria outlined in lines 77-79 re who eligible for recruitment

Response: We thank the reviewer for this feedback and have revised this statement accordingly. “Participants were eligible if they were active patients in one of the two HIV clinic study sites and had a self-reported diagnosis of hypertension as reflected in the criteria outlined below.” (page 6, lines 84-86)

7. Comment: Line 117 – don’t need to give range of age – just state median with the IQR

Response: We thank the reviewer for this comment and have revised this sentence to reflect the median age with IQR only. 

8. Comment: Line 119 – median time HIV disease – do you mean the time from HIV positive diagnosis? Bit worrying such a big gap between the two

Response: We thank the reviewer for this feedback. We have revised this phrase to convey the duration of time since HIV diagnosis. We acknowledge that there is a difference between duration of time since HIV diagnosis and duration of ART use. The WHO recommended initiation of ART regardless of CD4 level towards the end of 2015. As the median duration of time since HIV diagnosis was prior to this date, this likely accounts for gap in time between HIV diagnosis and ART use. 

9. Comment: Line 121- did the authors pick up where the hypertension diagnosis had been made if after the HIV diagnosis – in ART clinic- at another primary care clinic?

Response: We thank the reviewer for this thoughtful comment. We did identify where the hypertension diagnosis was made and have included these additional details in the results section. “No participant was diagnosed with hypertension in the HIV clinic; eight participants were diagnosed in the outpatient department (OPD), an urgent care clinic, two were diagnosed at a community health fair, one during a hospitalization, one in the preoperative setting, and one at a local pharmacy.” (page 8, paragraph 3)

10. Comment: Table 2: Listing lifestyle factors under poor understanding needs some explanation – were they not aware of diet, exercise etc

Response: We thank the reviewer for this feedback and have revised Table 2 to convey more detailed and descriptive themes and subthemes to provide further explanation. Please see our response to comments 8 and 9 from the editor. We have revised the subtheme ‘lifestyle factors’ to ‘lack of knowledge of some lifestyle factors such as sedentary lifestyle, obesity, smoking, and alcohol use as a cause’ (page 10, Table 2), and have added additional details in the text regarding knowledge of lifestyle factors. “All but 1 participant identified an unhealthy diet high in salt, fat and oil as a cause for hypertension. Few participants identified sedentary lifestyle and obesity as a cause. Only 1 participant identified alcohol use as a hypertension risk factor and no participant reported tobacco use as a cause. Chemicals and pesticides used in farming were also believed to be potential causes of hypertension.” (page 12, paragraph 2)

11. Comment: Line 370: edit queue

Response: We thank the reviewer for this comment and have revised this sentence to include the correct spelling of queue.

12. Comment: Line 373: may be worth putting USD equivalent

Response: We thank the reviewer for this detailed comment and have included the USD equivalent in order to provide broader context. 

13. Comment: Line 417 – agree the importance of health education but would also add on the importance placed on engaging people living with HIV in their care and the role of peer support to strengthen treatment literacy – this point I feel is also missing within the conclusion as a key message

Response: We thank the reviewer for this comment. We acknowledge the critical role of peer education in HIV outcomes and have added the following to the discussion to reflect this: “In addition, participants in our study reported mainly learning about hypertension from other members of the community, including hypertensive family members and friends. Given the success of peer education in HIV care in SSA [38–40] the role of peer educator in NCD management, including hypertension, should be strongly considered and further investigated.” (page 25, lines 536-540)

14. Comment: Line 459/60 – the authors may also like to reference the recommendation on screening for cardiovascular disease within the WHO 2016 guidelines for antiretroviral care

Do the authors know if there is a specific recommendation within the latest Tanzanian national guidelines for integration of ART and hypertension care - is the principle at least supported in national policy?

In the discussion on integration and service delivery the authors may also like to consider adding whether using the principles of differentiated service delivery- which Tanzania has adopted for HIV within their operational manual – to enhance hypertension care. I believe there is guidance within that document recommending NCD /HIV integration, so to highlight if policies are supported versus the challenges to fund and implement them

Response: We thank the reviewer for this thoughtful comment. We have reviewed the aforementioned WHO 2016 guidelines as well as the Tanzanian operation manual and have revised the discussion to reflect the key national policies that describe the importance of integrated care and further support the findings from our study. “Furthermore, the Tanzanian government has recognized the increased prevalence of NCDs among PLWH and the potential critical role of the HIV clinic. Recent national guidelines in Tanzania now support the integration of care for HIV and other diseases including hypertension and other NCDs [50], and an increasing number of studies in SSA suggest that the integration of HIV and NCD care is feasible, efficacious and cost effective [49,51,52]. (page 28, paragraph 1)

Reviewer #2: 

1. Comment: This is a very well written, clear manuscript which discusses the experiences of people living with HIV and hypertension in Tanzania, and reflects upon the challenges of managing this ‘dual burden’ of disease. Integration is an important issue to consider, especially in contexts with high HIV/NCD prevalence and where there are significant differences in the provision of medication (eg ARVs being free but hypertensive treatment not).

Response: We thank the reviewer for this positive feedback. We appreciate the detailed comments and feedback and have provided detailed responses below. 

2. Comment: There are limitations to the paper which are acknowledged by the authors, and which mostly relate to the study design. As participants self-reported their diagnosis of hypertension, there may have been recall bias or uncertainty around this. There is also a very small sample size – this is a qualitative study, so a small sample is expected, but the authors rely on data from only 13 patients making me question how thorough the analysis could be.

Response: We thank the reviewer for this thoughtful comment. We agree that both self-reported hypertension diagnosis and small sample size are major limitations to our study and have included both of these limitations to the discussion section. 

“This study had a few limitations. Firstly, inclusion criteria for hypertension was based on a self-reported diagnosis instead of a clinical diagnosis. Therefore, it is possible that some participants were not truly hypertensive, and thus fittingly did not engage in hypertension care. Second, social desirability bias when speaking with a non-Tanzanian physician may have influenced participants’ responses. Thirdly, our recruitment strategy excluded patients who had dropped out of HIV care. Therefore, we may have lost some additional insight into perspectives and experiences of an HIV-disengaged population. Fourthly, our recruitment strategy excluded patients without HIV; therefore, we were unable to compare how experiences of individuals with HIV and hypertension in our setting differ from those with hypertension alone. We also recognize that this is a relatively small qualitative sample size [54]. Although our analysis strategy identified data saturation by participant number 13, it is possible that recruitment from a larger sample may have yielded additional emerging themes.” (page 28, paragraph 2)

3. Comment: The perspectives of health-care providers are also missing, which would have been valuable in terms of making practical recommendations about how outcomes could be improved and understanding challenges from their perspective.

Response: We thank the reviewer for this comment. We agree that perspectives of health-care providers are also important and valuable in terms of making practical recommendations. We subsequently interviewed healthcare providers from three different facilities in northern Tanzania to explore their perspectives and experiences of hypertension care. The findings from the healthcare provider in-depth interviews will be discussed elsewhere in a future manuscript. 

4. Comment: Despite the limitations of the study design, the paper is well structured, the data well-presented and the discussion reflects upon many important, relevant issues to contexts within SSA, not just Tanzania.

Response: We thank the reviewer for this positive feedback. 

5. Comment: 

The manuscript focused mostly on experiences and perspectives and it is unclear what the ‘opportunities for improvement’ are eg clinical outcomes, cardiovascular outcomes, HIV outcomes, improvement in psycho-social well being…? I suggest removing ‘opportunities for improvement’ from the title. The rest of the manuscript refers to perspectives and experiences throughout, so I would edit the title to be in line with this.

Response: We thank the reviewer for this thoughtful comment. We agree that this paper focuses on perspectives and experiences and have revised the title accordingly to “It’s because I think too much”: Perspectives and experiences of adults with hypertension engaged in HIV care in northern Tanzania. 

6. Comment: Overall, the abstract is well written. I would like to see some additional figures about HIV and hypertension prevalence in Tanzania if the word limit allows.

Response: We appreciate the reviewer’s feedback. We have added the following additional statistics regarding HIV and hypertension prevalence in Tanzania in the abstract and in the introduction to provide further context and background to our study:

“In Tanzania, hypertension prevalence among PLWH approaches 20 to 30%. Despite this high burden of disease, most patients are unaware of their diagnosis and are not receiving treatment.” (page 2, paragraph 1)

 “SSA has the highest prevalence of hypertension in the world [9], and in Tanzania approximately 1 in 3 people may be hypertensive [10]. Despite this increased burden of disease, most patients with hypertension in SSA are not on treatment. A meta-analysis examining hypertension in SSA revealed that 73% of hypertensive adults were unaware of their diagnosis, 82% with a known diagnosis were not on treatment, and 93% of those on treatment had an uncontrolled blood pressure [11]. These concerning trends are also present among persons with HIV [12]: a recent study in northern Tanzania found that among a cohort of hypertensive patients enrolled in HIV care, 65% were unaware of their diagnosis, 90% were not on antihypertensive treatment and all had uncontrolled blood pressure [13].” (page 4, paragraph 2)

7. Comment: Line 52 – write out twofold in full

Response: We thank the reviewer for this detailed comment and have written out twofold in full as suggested. 

8. Comment: Line 53 – persons with HIV could be written in the standard PLWH which is used throughout

Response: We thank the reviewer for this comment and have revised the phrase ‘persons with HIV’ to PLWH as suggested. 

9. Comment: The methods section is very clear and detailed. COREQ guidelines are not mentioned – was the checklist adhered to and submitted, as per journal requirements?

Response: We thank the reviewer for this feedback. COREQ guidelines were adhered to. The guidelines have been described in the methods section as follows: “This qualitative study involved in-depth interviews with 13 individuals who were actively involved in HIV care and reported a history of hypertension. To ensure rigor and reproducibility, the presentation of methods and results follow the Consolidated Criteria for Reporting Qualitative Research (COREQ) guidelines (S1 File) [24].” (page 5, paragraph 3) The checklist has also been submitted as per journal requirements as a supporting document (S1 File. COREQ Checklist). 

10. Comment: Line 63 – as this is a qualitative paper, in line with other papers in the journal I suggest removing ‘materials’ as these are not relevant to this study

Response: We thank the reviewer for this feedback and have removed the word materials. 

11. Comment: Line 68 – why were these two clinics chosen? The abstract makes it sound like the study was conducted in ‘one of two’ clinics so I am unsure if it was one clinic, or both. Throughout the results, the clinics of the patients are not mentioned or linked with the quotes. Were there any differences between the two sites, or were they grouped together throughout rather than analysed separately?

Response: We thank the reviewer for this comment. We have revised the abstract to clarify that the study was conducted in two clinics. In addition, we have added a paragraph in the methods section to provide additional details about the two health facilities. Please see response to comment 5 from reviewer 1 for full details. We believe that there were no major differences between the two sites and therefore patients from the two sites were grouped together rather than analyzed separately. 

12. Comment: Line 73 – rather than recruitment being conducted until thematic saturation was reached, this should read ‘data collection was carried out until thematic saturation was reached’.

Response: We thank the reviewer for this feedback and have revised the statement to read ‘data collection was carried out until thematic saturation was reached’. 

13. Comment: Line 76 – who specifically were the HIV clinic and research staff? Was more than one person doing the recruitment at each site? A research assistant is mentioned later on, but it is unclear of how many different people were involved in the recruitment and data collection, and what their precise roles were.

Response: We thank the reviewer for this comment. Either the study nurse or a clinic nurse (one clinic nurse from each study site) assisted with recruitment. The physician researcher and the research assistant who also served as translator conducted all interviews and were involved in data collection. We have added the following details in the methods section to elaborate further on the study and clinic staff, recruitment and data collection: 

“During their routine HIV appointments patients were asked by the HIV clinic nurse or by the study nurse if they: 1) had a known diagnosis of hypertension, 2) were ever told by a health provider they had high blood pressure, and 3) had ever used medications to control blood pressure. If a patient met any of these criteria and also reported using ART for a minimum of 12 months, then the study nurse invited the patient to participate in the study.” (page 6, paragraph 3)

“All interviews were conducted in a private room at the study site by a female physician researcher from the US with graduate level training in qualitative methodology (PM), who was not involved in any clinical care at the study clinic sites. The interviews were conducted in English and were simultaneously translated in Swahili in real-time by a female Tanzanian research assistant (LM) who had received training in qualitative methodology.” (page 7, paragraph 1)

14. Comment: Line 78 – as mentioned by the authors, self-reported hypertension status is a limitation to the study. Would it have been possible to verify this through patient records?

Response: We thank the reviewer for this comment. While it may have been helpful to verify hypertension diagnosis through patient records, review of patient records was not included in the IRB or consent. In addition, patient records in the HIV clinic often include information pertaining to HIV only and not to other medical conditions and therefore review of records may not have provided additional information regarding hypertension status. 

15. Comment: Line 90 – the PI is referred to as a physician throughout, without mentioning their role as a researcher, or research training (in line with COREQ guidelines). Was the physician also a qualitative researcher? There are also other details missing from COREQ here (eg gender of interviewer/assistant).

Response: We thank the reviewer for this feedback. The physician received graduate level training in qualitative methodology. In line with the COREQ guidelines, we have added the following additional details to the methods section: “All interviews were conducted in a private room at the study site by a female physician researcher from the US with graduate level training in qualitative methodology (PM), who was not involved in any clinical care at the study clinic sites. The interviews were conducted in English and were simultaneously translated in Swahili in real-time by a female Tanzanian research assistant (LM) who had received training in qualitative methodology.” (page 7, paragraph 1)

16. Comment: Line 93 - I am very surprised that the interviews took so long as most of my interviews on similar issues do not take more than 30-45 minutes, so I would like to get some tips! Two hours is a very long time, especially when conducting interviews with people who – as the authors state – were often unsure of the treatment that they had received. Can the authors attach a copy of the interview guide in order to see the questions that were asked? As per COREQ guidelines, can you also state the mean duration and be more precise with interview length eg between 50 and 97 minutes.

Response: We thank the reviewer for this feedback. Interviews were conducted by the English speaker and every question and response was translated simultaneously in real-time by the research assistant in Swahili. This accounts for the length of the interviews. A copy of the interview guide has been attached as a supporting document (S2 File. In-depth Interview Guide). In addition, the mean and duration of interview length has been included in the methods section. “Interviews lasted 59 to 157 minutes with a median duration of 105 minutes, were audio recorded with participant consent, and were subsequently transcribed in English.” (page 7, lines 109-111) 

17. Comment: Line 94 – 3rd should be written in full

Response: We thank the reviewer for this comment and have written third in full. 

18. Comment: Line 95 – at what point were the audio-recordings translated? Was the transcription and translation carried out simultaneously by the person transcribing?

Response: We thank the reviewer for this comment. As previously described in the response to Comment 16, the research assistant translated every question and response in real time. A third member of the research team listened to 2 full audio recordings to confirm the on-site translation. The interviews were subsequently transcribed in full in English. 

19. Comment: Line 105 and prior - as there is a such a small number of interviews, I would give the number that were verified rather than the percentage (or both)

Response: We thank the reviewer for this comment and have provided both the number and percentage of interviews that were verified in the methods section. 

20. Comment: Line 114 - which language was the consent given in? Can the authors reflect upon the literacy levels in this population to explain why written rather than verbal consent was taken? This line should probably be rephrased to read ‘prior to data collection’ rather than ‘prior to enrolment’.

Response: We thank the reviewer for this comment. The consent was read out loud in Swahili by the research assistant and then the participant was asked to sign the consent. We have rephrased ‘prior to enrolment’ to ‘prior to data collection’. 

21. Comment: Line 118 – I suggest rephrasing ‘no more than a primary school education’ – I understand what the authors mean here, but the phrasing is a little judgemental.

Response: We thank the reviewer for this thoughtful feedback. We agree with the reviewer and have rephrased this sentence to state that most participants had some primary school education. 

22. Comment: Line 119 - would ‘median duration of HIV disease’ be ‘time since HIV diagnosis’ (see also table)? Is this information self-reported or from clinical records? (As an aside, if clinical records were consulted, was this mentioned in the consent, and who accessed them?)

Response: We thank the reviewer for this comment. Median duration of HIV disease was intended to convey the duration of time since HIV-diagnosis and we have clarified the phrasing to reflect this in the manuscript. Duration of time since HIV-diagnosis and duration of ART use were both self-reported. Clinical records were not included in the consent and therefore they were not accessed. 

23. Comment: Line 124 – 8 should read eight

Response: We thank the reviewer for this comment and have written eight in full. 

24. Comment: Table two, line 135 onwards – are some of the phrases under the themes ‘in vivo’ codes from NVivo, written exactly as the interviewees spoke them? It would be helpful to point these out, especially as some of them look grammatically incorrect in the context of a table.

Response: We thank the reviewer for this comment. We have revised Table 2 to communicate the themes and subthemes in a clearer manner and have quoted all phrases that were written exactly as the interviewees spoke them, i.e. “thinking too much”. 

25. Comment: I did not understand what ‘diagnosis in setting of symptoms’ meant -can you explain? It may be a clinical term I am unfamiliar with.

Response: We thank the reviewer for this comment. We have added additional details to Table 2 to clarify the meaning of ‘diagnosis in setting of symptoms’. We have rephrased the wording to ‘diagnosis made only after feeling ill and presenting with symptoms’, i.e. the participant was only diagnosed with hypertension after feeling ill and presenting to a health facility with symptoms. As hypertension is considered a silent disease and typically only manifests with symptoms long after complications and end-organ damage from uncontrolled hypertension have occurred, this is problematic. In addition, we have added the following to the results section to provide additional detail:

“Blood pressure was usually only measured when participants felt ill and presented to a health facility with symptoms, such as a headache, dizziness, fatigue or insomnia. All participants reported being diagnosed with hypertension when they presented with perceived symptoms. One interviewee described how she only checks her blood pressure when she feels unwell:” (page 16, paragraph 2)

26. Comment: Medications should be singular, not plural

Response: We thank the reviewer for this comment and have revised to medication. 

27. Comment: Overall, the data presented in the results in the form of quotes is well-written but I have some general comments on the presentation and introduction of the data. The quotes need more introduction eg ‘As explained by one of the interviewees below:’ or ‘As a 28 year old female interviewee stated:’ rather than just being listed. Not all the quotes link directly to the points being made prior to their introduction so some restructuring is required.

Response: We thank the reviewer for this feedback and have restructured all quotes and provided introductions as indicated. 

28. Comment: There are also some spacing and formatting issues which make the quotes difficult to read, and as PLoS One does not do copy editing, it would be helpful to check the formatting and allow more space between the quotes and the body of the text.

Response: We appreciate this thoughtful feedback. We have reformatted the quotes to allow for more space between quotes and the body of text. 

29. Comment: As there are a very small number of interviewees, I would prefer the authors to be more specific if they can. Whilst the aim of a qualitative paper is not to quantify responses (and I don’t want to suggest a full quantification of quotes at all), I feel in this case it would be beneficial to specify when it was only one participant, or all participants except for one etc rather than an over-reliance on many/few/some.

Response: We thank the reviewer for this comment. We have revised the results section to reflect the recommended changes and have specified if referring to only one participant versus all participants. As one example, on page 14, line 251, we have revised “Most respondents..” to “All but one respondent was able to identify some element of lifestyle modification that could improve blood pressure including reduction of salt, fat and oil.”

30. Comment: Line 163 - again, I understand what the authors mean here but rather than stating ‘poor levels’, is there a less judgemental way of phrasing this? Limited understanding, or lower levels, for example. I would also question whether calling high blood pressure ‘presha’ implies a poor level of knowledge, as suggested in the previous sentence, or is simply a very common Sub-Saharan African way of referring to it and thus implies an understanding of the condition.

Response: We appreciate this thoughtful comment and agree with the recommended suggestions. We have revised the phrase ‘poor levels of knowledge’ to ‘limited understanding’. In addition, we agree that the term presha does not imply a poor level of knowledge and it was not our intention to suggest so. Therefore, we have removed this sentence from the manuscript. 

31. Comment: Line 177 – this is interesting – can you provide more examples of those who talked about instantly dying or falling down?

Response: We appreciate this comment and as suggested have added the following quotes that represent the concept of instantly dying or falling down from hypertension:

“With pressure, if it goes up suddenly you will die, because I have seen that. That’s what happened to my mother. Suddenly her blood pressure became so high and she didn’t receive any help. The doctors later told us after the postmortem that due to high blood pressure some of the blood vessels near her neck burst and she died.” (Female, 45 years) (page 13, paragraph 2)

“It (hypertension) will kill you instantly. …I can give you an example of what happened to my neighbor. One day he just fell and when they went to help him they found that he had a stroke. And I asked, ‘What is stroke?’. They said, ‘Stroke is pressure’. And I said, ‘Oh no’. So high blood pressure can also do this to you.” (Female, 55 years) (page 13, paragraph 3)

32. Comment: Line 184 - this is a very long sentence, and the interesting story gets lost! Can you revise it?

Response: We thank the reviewer for this comment. We agree that this sentence could be more concisely and clearly stated and have revised the sentence as follows: “One participant who trained as a nursing assistant reported that she was taught in nursing school that stress is the most common cause of hypertension and believed that stress reduction could cure hypertension.” (page 13, lines 238-240)

33. Comment: Line 189 - should this be pharmacological treatment?

Response: We thank the reviewer for this suggestion and have changed the word pharmacologic to pharmacological. 

34. Comment: Line 194 – how was ‘noise’ translated? Can you refer back to the transcript? I wonder if this is about literal noise, or about avoiding stress or chaotic situations rather than noisy environments… This is an interesting point which I would like to understand more.

Response: We thank the reviewer for this comment. The word noise was about literal noise and avoiding noisy environments and we believe the interviewee may have been indicating a belief that noisy environments contribute to stress. We have clarified that this is environmental noise in the quote. 

35. Comment: Line 181 – how many were many? I am interested to know how many used religious or traditional treatments.

Response: We thank the reviewer for this comment. We have specified how many participants believed traditional, herbal and religious treatments could control hypertension and have revised the paragraph as follows: “All participants believed that hypertension could be controlled by reducing one’s worries. In addition, five participants believed that using herbal or traditional treatment could control blood pressure, three participants believed that temporary use of antihypertensive medication could cure hypertension, and two participants believed that hypertension could be cured through prayer.” (page 13, paragraph 5) 

36. Comment: Line 151 – the quote does not link back to the previous statement about side-effects

Response: We thank the reviewer for this comment and have restructured the quotes so that they directly link with corresponding statements in the text. 

37. Comment: Line 159 – should this be a few/some participants, rather than ‘few’?

Response: We thank the reviewer for this suggestion and have revised this sentence to specify the number of participants who believed lifestyle factors were a cause of hypertension. The paragraph now reads: “All but 1 participant identified an unhealthy diet high in salt, fat and oil as a cause for hypertension. Few participants identified sedentary lifestyle and obesity as a cause. Only 1 participant identified alcohol use as a hypertension risk factor and no participant reported tobacco use as a cause. Chemicals and pesticides used in farming were also believed to be potential causes of hypertension.” (page 12, paragraph 2)

38. Comment: Line 204 – how was this question asked? If it was a closed ended question, I could imagine most people would say yes to wanting additional information, but I am interested to know if people asked for extra information themselves, without prompting.

Response: We appreciate this comment. Participants were asked towards the end of the interview if they would like any additional information about hypertension. Due to social desirability bias it is possible that most participants would respond yes, and this has been included as a study limitation. However, there were a number of participants who asked unprompted questions about hypertension at different points throughout the interview and appeared interested in obtaining additional information for their health without any additional prompts. 

39. Comment: Line 207 – my concern with this section is that not everyone was on treatment, thus there is a smaller sample interviewed here. Can the authors clarify at this point how many people were on treatment, and how big the sample is in reference to pill burden. Similarly, on line 212, how many participants from the sample were asked this question, as presumably it wasn’t asked to those who were on treatment (which I think is only one interviewee).

Response: We thank the reviewer for this comment. All participants, including those who were not previously prescribed antihypertensive agents, were asked what their thoughts were about antihypertensive medication and how they would feel about taking antihypertensive medication chronically. These statements refer to all participants in our sample. 

40. Comment: Line 223 – this sentence is unclear to me. Do you mean challenges to adherence or interruption of treatment?

Response: We thank the reviewer for this comment. The phrase ‘no patient described adherence with hypertension care’ is referring to both adherence to antihypertensive medication as well as follow up with a medical provider about hypertension. The one participant who reported current use of antihypertensive agents admitted to non-adherence with one of his prescribed medications.

41. Comment: Line 226 – the word committed is somewhat problematic. Someone could be committed to taking their HIV treatment, but still be unable to do so because of financial means, stigma, unable to get to the clinic etc. The commitment is still there, but they are unable to carry out the action. The following quote does not link to the point about being ‘committed’ so I suggest the authors re-order this section.

Response: We appreciate the reviewer’s comment. We agree with the reviewer’s comment regarding the word committed and have removed the word committed and have revised this statement as follows: “Participants believed that there were alternative options to antihypertensive drugs for blood pressure control, whereas they believed the only way to control HIV was through ART.” (page 15, paragraph 4)

In addition, we have reordered all quotes to link directly with the corresponding text. 

42. Comment: Line 236 – the ‘setting of symptoms’ phrasing is unclear to me.

Response: We appreciate the reviewer’s comment. Please see our detailed response to comment number 25 regarding our revision and meaning of the phrase ‘diagnosis in the setting of symptoms’. 

43. Comment: Line 245 – including the age of this person would be helpful here – it is in the quote below, but could be moved up. The quote doesn’t mention the time period of several decades ago – can you include this to give more context?

Response: We thank the reviewer for this comment. We have introduced the quote with the participant’s age and also have provided additional detail about the time period in order to provide more context. 

44. Comment: Line 249 – OPD can be left in full as it is only used once in the manuscript

Response: We appreciate the reviewer’s comment. As the term OPD is used multiple times in the manuscript in the results and discussion, we have decided to include both the acronym and full phrase in the manuscript.

45. Comment: Line 257 – could this be rephrased as ‘challenges with provider communication and counselling’ ?

Response: We thank the reviewer for this thoughtful suggestion and have rephrased as challenges with provider communication and counseling. 

46. Comment: Line 262 – cited rather than ‘did cite’

Response: We appreciate this detailed feedback and have rephrased as cited. 

47. Comment: Line 264 – did this mean that they weren’t discussed regularly at follow up appointments, or that not all interviewees talked about monitoring?

Response: We appreciate the reviewer’s comment. Participants reported that healthcare providers did not discuss blood pressure monitoring or follow up plans and this statement has been revised as follows for clarification: “Participants reported that healthcare providers often did not discuss blood pressure monitoring and follow-up plans for hypertension. (page 17, lines 331-332) 

48. Comment: Line 272 – the below quote refers to arguments, rather than conflicts – did the longer audio-recording talk about more general conflict?

Response: We appreciate this comment. The audio recording discussed arguments with her husband, challenges within her marriage and marital strife and thus we chose to use the word conflict. 

49. Comment: Line 290 – rather than admitting reluctance, could this be rephrased to ‘were reluctant to use…’?

Response: We appreciate the reviewer’s feedback and have rephrased to were reluctant to use. 

50. Comment: Line 293 – the point about only being giving a prescription for 5-7 days is interesting, but could be more about practical reasons for not adhering to treatment (eg returning to the clinic weekly) than reluctance towards taking medication. Can the authors reflect on this?

Response: We thank the reviewer for this thoughtful feedback. We agree that a short duration of treatment had less to do with a reluctant attitude towards medication use but rather logistic challenges related to access to medication which may have led to nonadherence. A shorter duration of prescribed medication required frequent clinic visits which then created access issues related to limitations in transport, cost, and time which in turn likely resulted in issues with medication adherence. We have moved this statement to the Multiple barriers to hypertension care section and under the subtheme Lack of access to antihypertensive medication. 

51. Comment: Line 300 – I appreciate that this is a person’s words, but it would be easier to read if the grammar were corrected eg the doctor told me, rather than have told me. As long as none of the meaning is lost, a ‘tidying up’ of quotes throughout the manuscript would help the readers.

Response: We thank the reviewer for this thoughtful feedback. The quotes included in the results section were the original translation from the translator. We agree with the reviewer’s feedback and have revised all the quotes in the manuscript for readability. 

52. Comment: Line 308 – is the reference to ‘reduce their thinking’ a direct quote from someone? Is this referring to stress or something else?

Response: We thank the reviewer for this comment. ‘Reduce their thinking’ is a direct translated quote from all participants. However, the concept of ‘thinking too much’ is a commonly used idiom to reflect stress, anxiety and depression. Please see our response to comment 10 from the academic editor for additional details and revisions made to the manuscript to discuss the concept of ‘thinking too much’ in more detail. 

53. Comment: Line 315 – is traditional medicine man the term used locally, or does the more commonly used ‘traditional healer’ apply here?

Response: We thank the reviewer for this comment. The term traditional healer applies here, and we have rephrased this quote to convey this. 

54. Comment: Line 327 – which sub- category of ‘most’ are being discussed here? How many of the sample were not already in care – is this the same as the one person currently on treatment? How was this question phrased (eg open or closed question, or participants offered this information without prompting)?

Response: We appreciate the reviewer’s feedback. To quantify the number of participants and to clarify that we discussed hypertension follow up and treatment for all participants, we revised the sentence as follows: “Among the 13 participants, only 2 stated they had definitive plans to follow up with a medical provider for their hypertension care.” (page 20, line 408) This was asked as a closed ended question at times; however, some participants did offer this information without prompting. 

55. Comment: Line 330 – this is a very programmatic turn of phrase which feels out of place in the results! (Although I completely agree with the point). ‘Participants reported that HIV and hypertensive care were managed separately…’ could be one possible rephrasing.

Response: We thank the reviewer for this thoughtful feedback. We agree that this statement should not be included in the results and instead is better placed in the discussion. We have revised the statement to ‘Participants reported that HIV and hypertension care were managed separately’ as suggested. 

56. Comment: Line 343 – can you provide more examples here? This is only one patient, but you state that many stated a preference…

Response: We thank the reviewer for this comment and have provided the following additional quote to link to the text: “I would be so happy (to receive HIV and hypertension care in one location) because you may come today for one thing and a few days later another problem comes up. So, if they could check me for both things (at the same time) that would be really great.” (Female, 63 years) (page 22, paragraph 1)

57. Comment: Line 374 -this paragraph feels more like an analysis/reflection for the discussion, rather than part of the results. Can you rewrite it, or move to the discussion?

Response: We thank the reviewer for this comment and agree that this paragraph is better placed and reflected upon in the discussion. We have removed it from the results section and included the following in in the discussion for further reflection: “Our findings suggest that hypertension management was limited by inadequate provider counseling and communication. Participants reported minimal counseling from providers about hypertension, which may have led to low levels of knowledge about the disease. This limited awareness of the silent, chronic nature of hypertension and effective hypertension treatment likely contributed to misconceptions, including false attributions of hypertension cause, and subsequent disengagement and loss to follow up.” (page 26, paragraph 2)

58. Comment: This is very nicely written and the results, literature and reflections are woven together well. The limitations are very fair and well-reflected upon by the authors.

Response: We thank the reviewer for their positive feedback and very much appreciate their feedback and recommendations.

---

## [Editor Report · Decision Letter 1]

10 Aug 2020

PONE-D-20-04882R1

“It’s because I think too much”: Perspectives and experiences of adults with hypertension engaged in HIV care in northern Tanzania

PLOS ONE

Dear Dr. Manavalan,

Thank you for submitting your manuscript to PLOS ONE. After careful consideration, we feel that it has merit but does not fully meet PLOS ONE’s publication criteria as it currently stands. Therefore, we invite you to submit a revised version of the manuscript that addresses the points raised during the review process.

1. The paper has significantly improved.

2. The paper discusses a lot of general issues but could focus only on those related to PLHIV - paper could come down to 3,500 words.

3. Could also reduce quotes - maximum of two per point (and select the powerful ones).

4. Results could be re-arranged.

5. I am still struggling with some sections which mention lack of knowledge but later show participants had knowledge. Perhaps a more nuanced way of looking at this would be to say, 'When asked about what they were doing to reduce risk of hypertension, only 1 participant mentioned that they were taking less salt, fats, oil. However, when asked about the knowledge they had received from providers around this aspect, they mentioned several mitigation strategies, suggesting a mismatch between knowledge and practice' or something similar.

6. See additional suggestions in manuscript itself.

We look forward to receiving your revised manuscript.

Kind regards,

Webster Mavhu

Academic Editor

PLOS ONE

Additional Editor Comments (if provided):

1. The paper has significantly improved.

2. The paper discusses a lot of general issues but could focus only on those related to PLHIV - paper could come down to 3,500 words.

3. Could also reduce quotes - maximum of two per point (and select the powerful ones).

4. Results could be re-arranged.

5. I am still struggling with some sections which mention lack of knowledge but later show participants had knowledge. Perhaps a more nuanced way of looking at this would be to say, 'When asked about what they were doing to reduce risk of hypertension, only 1 participant mentioned that they were taking less salt, fats, oil. However, when asked about the knowledge they had received from providers around this aspect, they mentioned several mitigation strategies, suggesting a mismatch between knowledge and practice' or something similar.

6. See additional suggestions in manuscript itself.

---

## [Author Response · Author response to Decision Letter 1]

24 Sep 2020

The Editor

PLOS ONE

Re: PONE-D-20-04882, “It’s because I think too much”: Perspectives, experiences and opportunities for improvement among adults with hypertension engaged in HIV care in northern Tanzania

Dear Dr. Webster Mavhu,

Thank you again for the opportunity to provide a revision for this manuscript. We have addressed each comment and suggestion. The specific nature of the response to each point is outlined with our submission and changes are made, where appropriate, in the revised manuscript text. In particular, we have focused on issues specific to people living with HIV by revising the results and discussion to highlight the key barriers to hypertension care faced by people living with HIV. To our knowledge, there have been few qualitative manuscripts published in the literature exploring hypertension care experiences among people living with HIV in sub-Saharan Africa, and we believe our findings contribute to the literature supporting integrated HIV and chronic disease health services. 

This manuscript has not been previously published nor is it under consideration for publication elsewhere. The authors have seen and approved the version of the manuscript enclosed and concur with its submission to PLOS ONE. 

Thank you very much for considering this revision. 

Sincerely,

Preeti Manavalan, MD, MSc-GH

Assistant Professor of Medicine

Division of Infectious Diseases and Global Medicine

University of Florida

Academic Editor 

1. Comment: The paper has significantly improved. 

Response: We thank the editor for this positive feedback and have addressed specific comments and critical feedback below.

2. Comment: The paper discusses a lot of general issues but could focus only on those related to PLHIV - paper could come down to 3,500 words. 

Response: We thank the editor for this thoughtful feedback. We have carefully reviewed the manuscript with a view to tightening the text and focusing more on the issues unique to people living with HIV. We were able to reduce the text substantially, shortening from 6668 words to 4688 words. We felt that reducing the word count further may remove pertinent content and would delete content that reviewers had asked for in the initial review. The revised results now focus primarily on barriers experienced uniquely by people living with HIV. In addition, revisions to the discussion have been made to similarly focus on themes unique to people living with HIV (e.g., HIV-related stigma, burden from having multiple medical conditions, and siloed health care). We have also added the following paragraph to the discussion to highlight the significance of HIV-related stigma, stress and mental health; in doing so, we hope to make the added value of this study more apparent and bring more nuance to the paper. 

“Stress and HIV-related stigma were overarching themes described by all participants in our study. Participants believed that stress related to living with HIV caused hypertension. They also described feeling overwhelmed and “burdened” from having more than one medical condition. Moreover, some participants described experiences of HIV-related stigma which may have led to additional stress, anxiety, depression and social isolation. Chronic untreated mental health conditions and HIV-related stigma are well-known predictors of ART nonadherence and poor HIV clinical outcomes [39–41]. In addition, stigma avoidant behaviors, i.e. those actions patients take to avoid experiencing stigma such as concealing ART or non-disclosure of HIV status, may also contribute to nonadherence and disengagement with the medical system [42]. Furthermore, evidence suggests that chronic mental health conditions such as depression may also be a predictor for hypertension treatment nonadherence in similar settings [43]. Therefore, it is possible that psychosocial stressors, disease-related stigma and stigma-avoidant behaviors contributed to hypertension care disengagement and nonadherence in our sample. Further exploration of mental health conditions, stigma and their effects on clinical outcomes in HIV and other chronic diseases among PLHIV in SSA should be prioritized to better understand the factors contributing to care disengagement. In addition, due to stigmatization, patients with HIV may feel more comfortable navigating NCD care within the HIV clinical setting with providers in which an established and trusting relationship already exists. The effects of integrating health care on HIV-related stigma and mental health outcomes among PLHIV should also be further investigated.” (page 19, paragraph 2)

As we have substantially shortened this manuscript to focus on issues specific to people living with HIV, we have removed from the discussion reflections upon barriers that might be common across the general population of individuals with hypertension. 

3. Comment: Could also reduce quotes - maximum of two per point (and select the powerful ones).

Response: We thank the editor for this feedback. We have reduced the number of quotes in the results section and have selected quotes that focus on barriers unique to people living with HIV. 

4. Comment: Results could be re-arranged. 

Response: We thank the editor for this feedback. We have re-arranged the results so that the previous theme ‘Limited understanding of the causes of hypertension’ has been removed and is now discussed within the theme ‘Limited knowledge of hypertension’. In addition, we have revised the theme ‘Multiple barriers to hypertension care’ to ‘Additional structural barriers to hypertension care’ and have revised the subthemes within this theme to better reflect this change. The subthemes of ‘Additional structural barriers to hypertension care’ now include the following: High cost of care, Lack of access to antihypertensive medication, Staff shortages and long wait times, and HIV-associated stigma and social isolation.

5. Comment: I am still struggling with some sections which mention lack of knowledge, but later show participants had knowledge. Perhaps a more nuanced way of looking at this would be to say, 'When asked about what they were doing to reduce risk of hypertension, only 1 participant mentioned that they were taking less salt, fats, oil. However, when asked about the knowledge they had received from providers around this aspect, they mentioned several mitigation strategies, suggesting a mismatch between knowledge and practice' or something similar.

Response: We thank the editor for this comment. We have revised the results to reflect that knowledge about some aspects of hypertension were mixed and have added the following statement: “Knowledge regarding unhealthy lifestyle behaviors was mixed. All but one participant identified a diet high in salt and fat as a cause for hypertension. Few participants identified other unhealthy behaviors, including sedentary lifestyle, obesity, and alcohol and tobacco use as a cause.” (page 11, lines 181-184). In addition, as we have substantially shortened the manuscript to focus more on barriers specific to people living with HIV and less on barriers among the general population, we have removed from the results knowledge received from providers around lifestyle modification counseling.

6. Comment: See additional suggestions in manuscript itself.

Response: We thank the editor for the additional edits and suggestions in the manuscript itself. We have accepted the editor’s edits. We have included the comments in the track-changes version of the manuscript and included our responses as comments.

---

## [Editor Report · Decision Letter 2]

2 Oct 2020

PONE-D-20-04882R2

“It’s because I think too much”: Perspectives and experiences of adults with hypertension engaged in HIV care in northern Tanzania

PLOS ONE

Dear Dr. Manavalan,

We will accept the manuscript once the few suggested edits are addressed.

We look forward to receiving your revised manuscript.

Kind regards,

Webster Mavhu

Academic Editor

PLOS ONE

---

## [Author Response · Author response to Decision Letter 2]

12 Nov 2020

November 12, 2020

The Editor

PLOS ONE

Re: PONE-D-20-04882R2, “It’s because I think too much”: Perspectives and experiences of adults with hypertension engaged in HIV care in northern Tanzania

Dear Dr. Webster Mavhu,

Thank you again for the opportunity to provide a revision for this manuscript. We have addressed the few suggested edits and all changes have been made in the revised manuscript text. Please find below a detailed response to all comments. To our knowledge, there have been few qualitative manuscripts published in the literature exploring hypertension care experiences among people living with HIV in sub-Saharan Africa. We believe our findings contribute to the literature supporting integrated HIV and chronic disease health services, and may help address barriers to cardiovascular disease care among people living with HIV. Thank you very much for considering this revision. 

Sincerely,

Preeti Manavalan, MD, MSc-GH

Assistant Professor of Medicine

Division of Infectious Diseases and Global Medicine

University of Florida

Academic Editor 

1. Comment: We will accept the manuscript once the few suggested edits are addressed. 

Response: We thank the editor for this positive feedback and for the additional edits in the manuscript itself. We have accepted all of the editor’s suggestions and edits.

---

## [Editor Report · Decision Letter 3]

16 Nov 2020

“It’s because I think too much”: Perspectives and experiences of adults with hypertension engaged in HIV care in northern Tanzania

PONE-D-20-04882R3

Dear Dr. Manavalan,

We’re pleased to inform you that your manuscript has been judged scientifically suitable for publication and will be formally accepted for publication once it meets all outstanding technical requirements.

Kind regards,

Webster Mavhu

Academic Editor

PLOS ONE
---

## [Editor Report · Acceptance letter]

23 Nov 2020

PONE-D-20-04882R3 

“It’s because I think too much”: Perspectives and experiences of adults with hypertension engaged in HIV care in northern Tanzania 

Dear Dr. Manavalan:

I'm pleased to inform you that your manuscript has been deemed suitable for publication in PLOS ONE. Congratulations! Your manuscript is now with our production department. 

Kind regards, 

on behalf of

Dr. Webster Mavhu 

Academic Editor

PLOS ONE